# Active Diffusion Subsampling

**Oisín Nolan** *o.i.nolan@tue.nl*
**Tristan S.W. Stevens** *t.s.w.stevens@tue.nl*
**Wessel L. van Nierop** *w.l.v.nierop@tue.nl*
**Ruud J.G. van Sloun** *r.j.g.v.sloun@tue.nl*
*Eindhoven University of Technology*

Reviewed on OpenReview: *https://openreview.net/forum?id=OGifiton47*

## Abstract

Subsampling is commonly used to mitigate costs associated with data acquisition, such as time or energy requirements, motivating the development of algorithms for estimating the fully-sampled signal of interest $\mathbf{x}$ from partially observed measurements $\mathbf{y}$. In maximum-entropy sampling, one selects measurement locations that are expected to have the highest entropy, so as to minimize uncertainty about $\mathbf{x}$. This approach relies on an accurate model of the posterior distribution over future measurements, given the measurements observed so far. Recently, diffusion models have been shown to produce high-quality posterior samples of high-dimensional signals using *guided diffusion*. In this work, we propose *Active Diffusion Subsampling* (ADS), a method for designing intelligent subsampling masks using guided diffusion in which the model tracks a distribution of beliefs over the true state of $\mathbf{x}$ throughout the reverse diffusion process, progressively decreasing its uncertainty by actively choosing to acquire measurements with maximum expected entropy, ultimately producing the posterior distribution $p(\mathbf{x} \mid \mathbf{y})$. ADS can be applied using pre-trained diffusion models for any subsampling rate, and does not require task-specific retraining – just the specification of a measurement model. Furthermore, the maximum entropy sampling policy employed by ADS is interpretable, enhancing transparency relative to existing methods using black-box policies. Experimentally, we show that through designing informative subsampling masks, ADS significantly improves reconstruction quality compared to fixed sampling strategies on the MNIST and CelebA datasets, as measured by standard image quality metrics, including PSNR, SSIM, and LPIPS. Furthermore, on the task of Magnetic Resonance Imaging acceleration, we find that ADS performs competitively with existing supervised methods in reconstruction quality while using a more interpretable acquisition scheme design procedure. Code is available at `https://active-diffusion-subsampling.github.io/`.

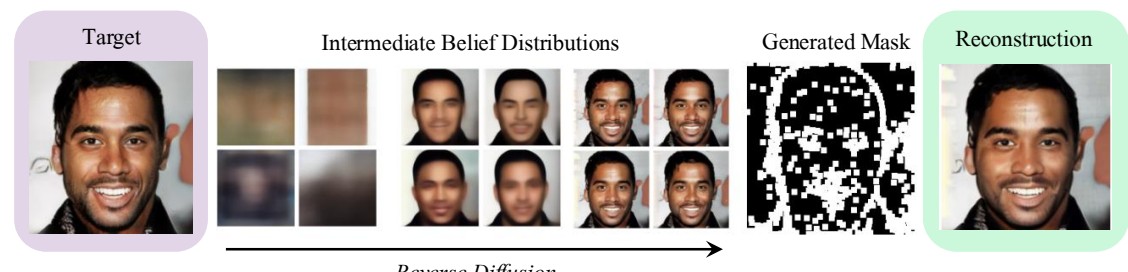

Figure 1: *Active Diffusion Subsampling* jointly designs a subsampling mask and reconstructs the target signal in a single reverse diffusion process.

# 1 Introduction

In recent years, diffusion models have defined the state of the art in inverse problem solving, particularly in the image domain, through novel posterior sampling methods such as *Diffusion Posterior Sampling* (DPS) (Chung et al., 2022) and *Posterior Sampling with Latent Diffusion* (PSLD) (Rout et al., 2024). These methods are often evaluated on inverse imaging problems, such as inpainting, which is akin to image subsampling. Typical benchmarks evaluate inpainting ability using naïve subsampling masks such as randomly masked pixels, or a box mask in the center of the image (Rout et al., 2024). In real-world applications, however, more sophisticated subsampling strategies are typically employed, for example, in Magnetic Resonance Imaging (MRI) acceleration (Lustig & Pauly, 2010; Bridson, 2007). These subsampling strategies are usually designed by domain experts, and are therefore not generalizable across tasks. Some recent literature has explored learning subsampling masks for various tasks (Bahadir et al., 2020; Baldassarre et al., 2016; Huijben et al., 2020; Van Gorp et al., 2021), but these methods typically depend on black-box policy functions, and require task-specific training. In this work, we introduce *Active Diffusion Subsampling* (ADS), an algorithm for automatically designing task- and sample-adaptive subsampling masks using diffusion models, without the need for further training or fine-tuning (Figure 2). ADS uses a white-box policy function based on maximum entropy sampling (Caticha, 2021), in which the model chooses sampling locations that are expected to maximize the information gained about the reconstruction target. In order to implement this policy, ADS leverages quantities that are already computed during the reverse diffusion process, leading to minimal additional inference time. We anticipate that ADS can be employed by practitioners in various domains that use diffusion models for subsampling tasks but currently rely on non-adaptive subsampling strategies. Such domains include medical imaging, such as MRI, CT, and ultrasound (Shaul et al., 2020; Pérez et al., 2020; Faltin et al., 2011), geophysics (Campman et al., 2017; Cao et al., 2011), and astronomy (Feng et al., 2023). Our main contributions are thus as follows:

- A novel approach to active subsampling which can be employed with existing diffusion models using popular posterior sampling methods;

- A white-box policy function for sample selection, grounded in theory from Bayesian experimental design;

- Significant improvement in reconstruction quality relative to baseline sampling strategies.

- Application to MRI acceleration and ultrasound scan-line subsampling.

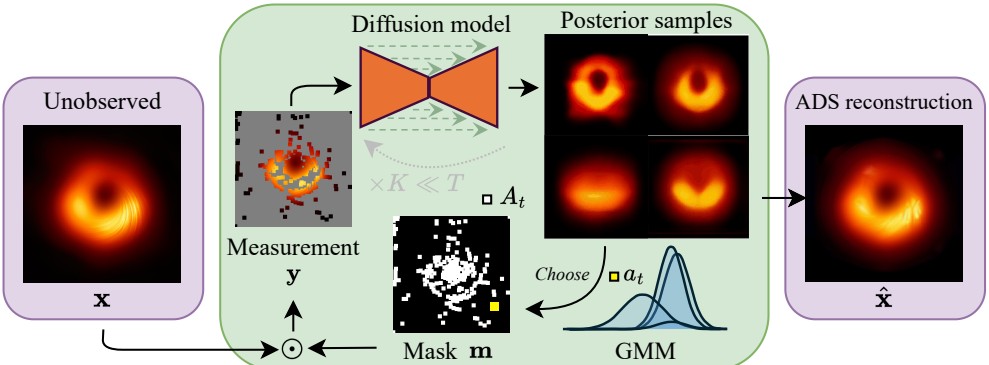

Figure 2: Schematic overview of the proposed Active Diffusion Sampling (ADS) method.

# 2 Related Work

Methods aiming to select maximally informative measurements appear in many domains, spanning statistics, signal processing, and machine learning, but sharing foundations in information theory and Bayesian

inference. Optimal Bayesian Experimental Design (Lindley, 1956) aims to determine which experiment will be most informative about some quantity of interest $\theta$ (Rainforth et al., 2024), typically parameters of a statistical model. Active learning (Houlsby et al., 2011) performs an analogous task in machine learning, aiming to identify which samples, if included in the training set, would lead to the greatest performance gain on the true data distribution. While our method focuses on subsampling high-dimensional signals, it could also be interpreted as a Bayesian regression solver for the forward problem $\mathbf{y} = f(\mathbf{x}) + \mathbf{n}$, in which $\mathbf{x}$ is now seen as parameters for a model $f$, and the active sampler seeks measurements $\mathbf{y}$ which minimize uncertainty about the parameters $\mathbf{x}$, relating it to the task of Bayesian experimental design.

From a signal processing perspective, ADS can be characterized as a novel approach to adaptive compressive sensing, in which sparse signals are reconstructed from measurements with sub-Nyquist sampling rates (Rani et al., 2018), and typically applied in imaging and communication. Measurement matrices relating observed measurements to the signal of interest $\mathbf{x}$ are then designed so as to minimize reconstruction error on $\mathbf{x}$. In Bayesian compressive sensing, measurement matrices are designed so as to minimize a measure of uncertainty about the value of $\mathbf{x}$. Adaptive approaches, such as that of Braun et al. (2015), aim to decrease this uncertainty iteratively, by greedily choosing measurements that maximize the mutual information between $\mathbf{x}$ and $\mathbf{y} = \mathbf{Ux} + \mathbf{n}$. These methods typically assume a Gaussian prior on $\mathbf{x}$, however, limiting the degree to which more complex prior structure can be used. More recently, a number of methods using deep learning to design subsampling strategies have emerged. These approaches typically learn subsampling strategies from data that minimize reconstruction error between $\mathbf{x}$ and $\mathbf{y}$. Methods by Huijben et al. (2020) and Bahadir et al. (2020) learn fixed sampling strategies, in which a single mask is designed *a priori* for a given domain, and applied to all samples for inference. These methods can be effective, but suffer in cases where optimal masks differ across samples. Sample-adaptive methods (Van Gorp et al., 2021; Bakker et al., 2020; Yin et al., 2021; Stevens et al., 2022) move past this limitation by designing sampling strategies at inference time. A popular application of such methods is MRI acceleration, spurred by the fastMRI benchmark (Zbontar et al., 2018), in which a full MRI image must be reconstructed from sub-sampled $\kappa$-space measurements. In A-DPS (Van Gorp et al., 2021), for example, a neural network is trained to build an acquisition mask consisting of $M$ $\kappa$-space lines iteratively, adaptively adding new lines based on the current reconstruction and prior context. Bakker et al. (2020) implements the same procedure using a reinforcement learning agent. One drawback of these methods is their reliance on black-box policies, making it difficult to detect and interpret failure cases. Generative approaches with transparent sampling policies circumvent this issue. For example, methods by Sanchez et al. (2020) and van de Camp et al. (2023) take a generative approach to adaptive MRI acquisition design, using variants of generative adversarial networks (GANs) to generate posterior samples over MRI images, and maximum-variance sampling in the $\kappa$-space as the measurement selection policy. The performance of such generative approaches depends on how well they can model the true posterior distribution over $\mathbf{x}$ given observations. This motivates our choice of diffusion models for generative modeling, as they have shown excellent performance in diverse domains, such as computer vision (Dhariwal & Nichol, 2021), medical imaging (Chung & Ye, 2022; Stevens et al., 2024), and natural language processing (Yu et al., 2022). The existing works discussed above have in common that they run the reconstruction model $N$ times for $N$ measurements during inference. Due to the slower iterative inference procedure used by diffusion models, running inference $N$ times may not be feasible in real-world applications, which are typically time-bound by the acquisition process. ADS alleviates this issue by using the Tweedie estimate of the posterior distribution as an approximate posterior, allowing $N$ measurements to be acquired in a single reverse diffusion process. We also highlight a related concurrent work by Elata et al. (2025), which uses denoising diffusion restoration models (DDRMs) to generate sensing matrices for adaptive compressed sensing, which differs from ADS in that it requires full DDRM inference in order to acquire each sample.

## 3 Background

### 3.1 Bayesian Optimal Experimental Design

In Bayesian Optimal Experimental Design (Rainforth et al., 2024) and Bayesian Active Learning (Houlsby et al., 2011), the objective is to choose the optimal *design*, or set of actions $A = A^*$, leading to new observations of a measurement variable $\mathbf{y}$ that will minimize uncertainty about a related quantity of interest

$\mathbf{x}$, as measured by entropy $H$. It was shown by Lindley (1956) that this objective is equivalent to finding the actions $A$ that maximize the mutual information $I$ between $\mathbf{x}$ and $\mathbf{y}$, i.e. selecting actions leading to observations of $\mathbf{y}$ that will be most informative about $\mathbf{x}$:

$$A^* = \arg\min_A[H(\mathbf{x} \mid A, \mathbf{y})] = \arg\max_A[I(\mathbf{y}; \mathbf{x} \mid A)] \tag{1}$$

This objective is commonly optimized *actively*, wherein the design is created iteratively by choosing actions that maximize mutual information, considering past observations when selecting new actions (Rainforth et al., 2024). This active paradigm invites an agent-based framing, in which the agent's goal is to minimize its own uncertainty, and beliefs over $\mathbf{x}$ are updated as new measurements of $\mathbf{y}$ are taken. The active design objective can be formulated as follows, where $a_t$ is possible action at time $t$, $A_t = A_{t-1} \cup a_t$ is the set of actions $\{a_0, ..., a_t\}$ taken so far at time $t$, and $\mathbf{y}_{t-1}$ is the set of partial observations of the measurement variable $\mathbf{y}$ until time $t-1$:

$$
\begin{aligned}
a_t^* &= \arg\max_{a_t}[I(\mathbf{y}_t; \mathbf{x} \mid A_t, \mathbf{y}_{t-1})] \\
&= \arg\max_{a_t}[\mathbb{E}_{p(\mathbf{y}_t|\mathbf{x},A_t)p(\mathbf{x}|y_{t-1})}[\log p(\mathbf{y}_t \mid A_t, \mathbf{y}_{t-1}) - \log p(\mathbf{y}_t \mid \mathbf{x}, A_t, \mathbf{y}_{t-1})]] \\
&= \arg\max_{a_t}[H(\mathbf{y}_t \mid A_t, \mathbf{y}_{t-1}) - H(\mathbf{y}_t \mid \mathbf{x}, A_t, \mathbf{y}_{t-1})]
\end{aligned}
\tag{2}
$$

We can interpret the agent's behavior from Equation 2 as trying to maximize marginal uncertainty about $\mathbf{y}_t$, while minimizing model uncertainty about what value $\mathbf{y}_t$ should take on given a particular $\mathbf{x}$ (Houlsby et al., 2011). Active designs are typically preferred over *fixed designs*, in which a set of actions is chosen up-front, as opposed to being chosen progressively as measurements are acquired (Rainforth et al., 2024). While fixed designs may be more computationally efficient, they are less sample-specific, which can lead to lower information gain about $\mathbf{x}$. Finally, it is worth noting that this active optimization scheme, while greedy, has been shown to be near-optimal due to the submodularity of conditional entropy (Golovin & Krause, 2011).

## 3.2 Subsampling

Generally image reconstruction tasks can be formulated as inverse problems, given by:

$$\mathbf{y} = \mathbf{U}\mathbf{x} + \mathbf{n}, \tag{3}$$

where $\mathbf{y} \in \mathcal{Y}^M$ is a measurement, $\mathbf{x} \in \mathcal{X}^N$ the signal of interest and $\mathbf{n} \in \mathcal{N}^M$ some noise source, typically Gaussian. For the subsampling problem, the measurement matrix $\mathbf{U} \in \mathbb{R}^{M \times N}$ can be expressed in terms of a binary subsampling matrix using one-hot encoded rows such that we have an element-wise mask $\mathbf{m} = \text{diag}(\mathbf{U}^\top \mathbf{U})$, where only the diagonal entries of $\mathbf{U}^\top \mathbf{U}$ are retained, representing the subsampling pattern. We can relate the subsampling mask $\mathbf{m}$ through the zero-filled measurement which can be obtained through $\mathbf{y}_{\text{zf}} = \mathbf{U}^\top \mathbf{y} = \mathbf{m} \odot \mathbf{x} + \mathbf{U}^\top \mathbf{n}$.

Since we are interested in the adaptive design of these masks, we express their generation as $\mathbf{m} = \mathbf{U}(A_t)^\top \mathbf{U}(A_t)$, where the measurement matrix is now a function of the actions $A_t = \{a_0, ..., a_t\}$ taken by the agent up to time $t$. The $i^{\text{th}}$ element of that mask is defined as follows:

$$\mathbf{m} = \left[\mathbf{U}(A_t)^\top \mathbf{U}(A_t)\right]_i = \begin{cases} 1 & \text{if } i \in A_t \\ 0 & \text{otherwise.} \end{cases} \tag{4}$$

The measurement model in equation 3 can now be extended to an active setting via $\mathbf{y}_t = \mathbf{U}(A_t)\mathbf{x} + \mathbf{n}_t$. Note too that in some applications we have an additional *forward model* $f$, mapping from the data domain to the measurement domain, yielding $\mathbf{y}_t = \mathbf{U}(A_t)f(\mathbf{x}) + \mathbf{n}_t$.

## 3.3 Posterior Sampling with Diffusion Models

Denoising diffusion models learn to reverse a stochastic differential equation (SDE) that progressively noises samples $\mathbf{x}$ towards a standard Normal distribution (Song et al., 2020). The (variance preserving) SDE

defining the noising process is as follows:

$$d\mathbf{x} = -\frac{\beta(\tau)}{2}\mathbf{x}d\tau + \sqrt{\beta(\tau)}d\mathbf{w} \tag{5}$$

where $\mathbf{x}(0) \in \mathbb{R}^d$ is an initial clean sample, $\tau \in [0, T]$, $\beta(\tau)$ is the noise schedule, and $w$ is a standard Wiener process, and $\mathbf{x}(T) \sim \mathcal{N}(0, I)$. According to the following equation from Anderson (1982), this SDE can be reversed once the score function $\nabla_\mathbf{x} \log p_\tau(\mathbf{x})$ is known, where $\bar{w}$ is a standard Wiener process running backwards:

$$d\mathbf{x} = \left[-\frac{\beta(\tau)}{2}\mathbf{x} - \beta(\tau)\nabla_\mathbf{x} \log p_\tau(\mathbf{x})\right]d\tau + \sqrt{\beta(\tau)}d\bar{\mathbf{w}} \tag{6}$$

Following the notation by Ho et al. (2020) and Chung et al. (2022), the discrete setting of the SDE is represented using $x_\tau = \mathbf{x}(\tau T/N), \beta_\tau = \beta(\tau T/N), \alpha_\tau = 1 - \beta_\tau, \bar{\alpha}_\tau = \prod_{s=1}^\tau \alpha_s$, where $N$ is the numbers of discretized segments. The diffusion model achieves the SDE reversal by learning the score function using a neural network parameterized by $\theta$, $s_\theta(\mathbf{x}_\tau, \tau) \simeq \nabla_{\mathbf{x}_\tau} \log p_\tau(\mathbf{x}_\tau)$.

The reverse diffusion process can be conditioned on a measurement $\mathbf{y}$ to produce samples from the posterior $p(\mathbf{x}|\mathbf{y})$. This can be done with substitution of the *conditional score function* $\nabla_{\mathbf{x}_\tau} \log p_\tau(\mathbf{x}_\tau)$ in equation 6. The intractability of the noise-perturbed likelihood $\nabla_{\mathbf{x}_\tau} \log p_\tau(\mathbf{y}|\mathbf{x}_\tau)$ which follows from refactoring the posterior using Bayes' rule has led to various approximate guidance schemes to compute these gradients with respect to a partially-noised sample $\mathbf{x}_\tau$ (Chung et al., 2022; Song et al., 2023; Rout et al., 2024). Most of these rely on Tweedie's formula (Efron, 2011; Robbins, 1992), which serves as a one-step denoising process from $\tau \to 0$, denoted $\mathcal{D}_\tau(.)$, yielding the Minimum Mean-Squared Error denoising of $\mathbf{x}_\tau$ (Milanfar & Delbracio, 2024). Using our trained score function $s_\theta(\mathbf{x}_\tau, \tau)$ we can approximate $\mathbf{x}_0$ as follows:

$$\hat{\mathbf{x}}_0 = \mathbb{E}[\mathbf{x}_0|\mathbf{x}_\tau] = \mathcal{D}_\tau(\mathbf{x}_\tau) = \frac{1}{\sqrt{\bar{\alpha}_\tau}}(\mathbf{x}_\tau + (1 - \bar{\alpha}_\tau)s_\theta(\mathbf{x}_\tau, \tau)) \tag{7}$$

*Diffusion Posterior Sampling* (DPS) uses equation 7 to approximate $\nabla_{\mathbf{x}_\tau} \log p(\mathbf{y}|\mathbf{x}_\tau) \approx \nabla_{\mathbf{x}_\tau} \log p(\mathbf{y}|\hat{\mathbf{x}}_0)$. In the case of active subsampling, this leads to guidance term $\nabla_{\mathbf{x}_\tau} ||\mathbf{y}_t - \mathbf{U}(A_t)f(\hat{\mathbf{x}}_0)||_2^2$ indicating the direction in which $\mathbf{x}_\tau$ should step in order to be more consistent with $\mathbf{y}_t = \mathbf{U}(A_t)f(\mathbf{x}) + \mathbf{n}$. The conditional reverse diffusion process then alternates between standard reverse diffusion steps and guidance steps in order to generate samples from the posterior $p(\mathbf{x} \mid \mathbf{y}_t)$. Finally, we note that a number of posterior sampling methods for diffusion models have recently emerged, and refer the interested reader to the survey by Daras et al. (2024).

## 4 Method

### 4.1 Active Diffusion Subsampling

ADS (Algorithm 1) operates by running a reverse diffusion process generating a batch $\{\mathbf{x}_\tau^{(i)}\}, i \in 0, ..., N_p$ of possible reconstructions of the target signal guided by an evolving set of measurements $\{\mathbf{y}_t\}_{t=0}^{\mathcal{T}}$ which are revealed to it through subsampling actions taken at reverse diffusion steps satisfying $\tau \in S$, where $S$ is a *subsampling schedule*, or set of diffusion time steps at which to acquire new measurements. For example, in an application in which one would like to acquire 10 measurements, $\{\mathbf{y}_t\}_{t=0}^9$, across a diffusion process consisting of $T = 100$ steps, one might choose a linear subsampling schedule $S = \{0, 10, 20, ..., 90\}$. Then, during the diffusion process, if the diffusion step $\tau$ is an element of $S$, a new measurement will be acquired, revealing it to the diffusion model. These measurements could be, for example, individual pixels or groups of pixels. We refer to the elements of the batch of reconstructions $\{\mathbf{x}_\tau^{(i)}\}$ as *particles* in the data space, as they implicitly track a belief distribution over the true, fully-denoised $\mathbf{x} = \mathbf{x}_0$ throughout reverse diffusion. These particles are used to compute estimates of uncertainty about $\mathbf{x}$, which ADS aims to minimize by choosing actions $a_t$ that maximize the mutual information between $\mathbf{x}$ and $\mathbf{y}_t$ given $a_t$. Figure 3 illustrates this action selection procedure. The remainder of the section describes how this is achieved through (i) employing running estimates of $\mathbf{x}_0$ given by $\mathcal{D}_\tau(\mathbf{x}_\tau)$, (ii) modeling assumptions on the measurement entropy, and (iii) computational advantages afforded by the subsampling operator.

---

**Algorithm 1:** Active Diffusion Subsampling

    **Require:** $T, N_p, S, \zeta, \{\tilde{\sigma}_\tau\}_{\tau=0}^T, \{\alpha_\tau\}_{\tau=0}^T, A_{\texttt{init}}$

**1**   $t = 0; \quad A_0 = A_{\texttt{init}}; \quad \mathbf{y}_0 = \mathbf{U}(A_0)f(\mathbf{x}) + \mathbf{n}_0; \quad \{\mathbf{x}_T^{(i)} \sim \mathcal{N}(\mathbf{0}, \mathbf{I})\}_{i=0}^{N_p-1}$

**2**   **for** $\tau = T$ **to** $0$ **do**

      // Batch process in parallel for efficient inference

**3**      **for** $i = 0$ **to** $N_p - 1$ **do**

**4**          $\hat{\mathbf{s}} \leftarrow \mathbf{s}_\theta(\mathbf{x}_\tau^{(i)}, \tau)$

**5**          $\hat{\mathbf{x}}_0^{(i)} \leftarrow \mathcal{D}_\tau(\mathbf{x}_\tau^{(i)}) = \frac{1}{\sqrt{\bar{\alpha}_\tau}}(\mathbf{x}_\tau^{(i)} + (1 - \bar{\alpha}_\tau)\hat{\mathbf{s}})$ // Calculate Tweedie estimate

**6**          $\hat{\mathbf{y}}^{(i)} \leftarrow f(\hat{\mathbf{x}}_0^{(i)})$ // Simulate full measurement

**7**          $\mathbf{z} \sim \mathcal{N}(\mathbf{0}, \mathbf{I})$

**8**          $\mathbf{x}_{\tau-1}^{(i)\prime} \leftarrow \frac{\sqrt{\alpha_\tau}(1-\bar{\alpha}_{\tau-1})}{1-\bar{\alpha}_\tau}\mathbf{x}_\tau^{(i)} + \frac{\sqrt{\bar{\alpha}_{\tau-1}}\beta_\tau}{1-\bar{\alpha}_\tau}\hat{\mathbf{x}}_0^{(i)} + \tilde{\sigma}_\tau \mathbf{z}$ // Reverse diffusion step

**9**          $\mathbf{x}_{\tau-1}^{(i)} \leftarrow \mathbf{x}_{\tau-1}^{(i)\prime} - \zeta\nabla_{\mathbf{x}_\tau^{(i)}}||\mathbf{y}_t - \mathbf{U}(A_t)\hat{\mathbf{y}}^{(i)}||_2^2$ // Data fidelity step

**10**      **if** $\tau \in S$ **then**

**11**          $t \leftarrow t + 1$

        // Find the group of pixels with maximum entropy across simulated measurements

**12**          $a_t^* = \arg\max_{a_t}\left[\sum_i^{N_p} \log \sum_j^{N_p} \exp\left\{\frac{\sum_{l\in a_t}(\hat{\mathbf{y}}_l^{(i)} - \hat{\mathbf{y}}_l^{(j)})^2}{2\sigma_{\mathbf{y}}^2}\right\}\right]$

**13**          $A_t = A_{t-1} \cup a_t^*$

**14**          $\mathbf{y}_t = \mathbf{U}(A_t)f(\mathbf{x}) + \mathbf{n}_t$ // Acquire new measurements of the true x

**15**      **end**

    **return:** $\{\hat{\mathbf{x}}_0^{(i)}\}_{i=0}^{N_p-1}$ // Return posterior samples

---

ADS follows an information-maximizing policy, selecting measurements $a_t^* = \arg\max_{a_t}[I(\mathbf{y}; \mathbf{x} \mid A_t, \mathbf{y}_{t-1})]$. Assuming a measurement model with additive noise $\mathbf{y}_t = \mathbf{U}(A_t)f(\mathbf{x}) + \mathbf{n}_t$, the posterior entropy term $H(\mathbf{y}_t \mid \mathbf{x}, A_t, \mathbf{y}_{t-1})$ in the mutual information is unaffected by the choice of action $a_t$: it is solely determined by the noise component $\mathbf{n}_t$. This simplifies the policy function, leaving only the marginal entropy term $H(\mathbf{y}_t \mid A_t, \mathbf{y}_{t-1})$ to maximize. This entropy term (Equation 2) can be computed as the expectation of $\log p(\mathbf{y}_t \mid A_t, \mathbf{y}_{t-1})$, or the expected distribution of measurements $\mathbf{y}_t$ given the observations so far, and a subsampling matrix $A_t$. We approximate the true measurement posterior in Equation 8 as a mixture of $N_p$ isotropic Gaussians, with means set as estimates of future measurements under possible actions, and variance $\sigma_y^2\mathbf{I}$. The means $\hat{\mathbf{y}}_t^{(i)} = \mathbf{U}(A_t)f(\hat{\mathbf{x}}_0^{(i)})$ are computed by applying the forward model to posterior samples $\hat{\mathbf{x}}_0^{(i)} \sim p(\mathbf{x} \mid \mathbf{y}_{t-1})$, which are estimated using the batch of partially denoised particles $\mathbf{x}_\tau^{(i)}$ via $\hat{\mathbf{x}}_0^{(i)} = \mathcal{D}_\tau(\mathbf{x}_\tau^{(i)})$, yielding:

$$p(\mathbf{y}_t \mid A_t, \mathbf{y}_{t-1}) = \sum_{i=0}^{N_p} w_i \mathcal{N}(\hat{\mathbf{y}}_t^{(i)}, \sigma_{\mathbf{y}}^2\mathbf{I}) \tag{8}$$

Note the parameter $\sigma_y^2$, which controls the width of the Gaussians . The entropy of this isotropic Gaussian Mixture Model can then be approximated as follows, as given by Hershey & Olsen (2007):

$$H(\mathbf{y}_t \mid A_t, \mathbf{y}_{t-1}) \approx \text{constant} + \sum_i^{N_p} w_i \log \sum_j^{N_p} w_j \exp\left\{\frac{||\hat{\mathbf{y}}_t^{(i)} - \hat{\mathbf{y}}_t^{(j)}||_2^2}{2\sigma_{\mathbf{y}}^2}\right\} \tag{9}$$

Intuitively, this approximation computes the entropy of the mixture of Gaussians by summing the Gaussian error between each pair of means, which grows with distance according to the hyperparameter $\sigma_{\mathbf{y}}^2$. This hyperparameter $\sigma_{\mathbf{y}}^2$ can therefore be tuned to control the sensitivity of the entropy to outliers.

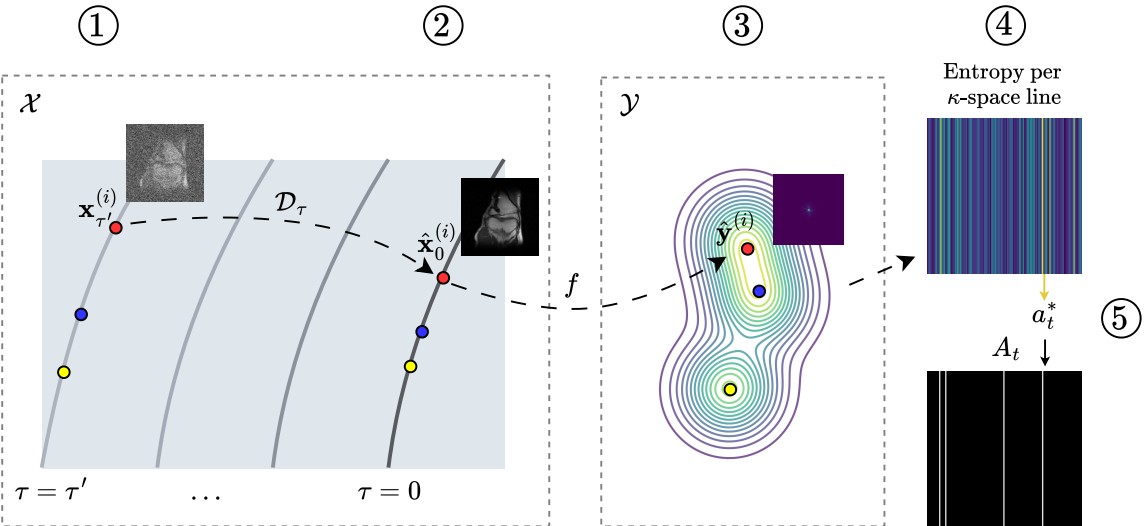

Figure 3: Illustration of a single action selection using ADS. ① shows the current batch of partially-denoised images $\{\mathbf{x}_{\tau'}^{(i)}\}$ at diffusion step $\tau'$. In ②, this batch of particles is mapped using Tweedie's formula to a batch of fully-denoised particles $\{\hat{\mathbf{x}}_0^{(i)}\}$, constituting the belief distribution at time $\tau'$. The forward model $f$ is then applied in ③ to simulate the set of measurements that result from the belief distribution, used to approximate the measurement posterior as a GMM. Given this GMM, the measurement entropy is computed in ④ using Equation 10. Finally, in ⑤, the maximum entropy line is selected as the next measurement location.

Finally, we leverage the fact that $\mathbf{U}(A_t)$ is a subsampling matrix to derive an efficient final formulation of our policy. Because $\mathbf{U}(A_t)$ is a subsampling matrix, the optimal choice for the next action $a_t$ will be at the region of the measurement space with the largest disagreement among the particles, as measured by Gaussian error in Equation 9. Therefore, rather than computing a separate set of subsampled particles for each possible next subsampling mask, we instead compute a single set of fully-sampled measurement particles $\hat{\mathbf{y}}^{(i)} = f(\hat{\mathbf{x}}_0^{(i)})$, and simply choose the optimal action as the region with largest error. For example, when pixel-subsampling an image, the particles $\hat{\mathbf{y}}^{(i)}$ become predicted estimates of the full image, given the pixels observed so far, and the next sample is chosen as whichever pixel has the largest total error across the particles. Similarly, in accelerated MRI, the next $\kappa$-space line selected is the one in which there is the largest error across estimates of the full $\kappa$-space. Denoting as $l \in a_t$ the set of indices sampled by each possible action $a_t$, and assuming equal weights for all particles, $w_i = w_j, \forall i, j$, the final form of the policy function is given as follows (see Appendix A.1 for derivation):

$$a_t^* = \arg\max_{a_t} \left[ \sum_i^{N_p} \log \sum_j^{N_p} \exp \left\{ \frac{\sum_{l \in a_t} (\hat{\mathbf{y}}_l^{(i)} - \hat{\mathbf{y}}_l^{(j)})^2}{2\sigma_y^2} \right\} \right] \tag{10}$$

## 5 Experiments

We evaluate our method with three sets of experiments, covering a variety of data distributions and application domains. The first two experiments, in Sections 5.1 and 5.2, evaluate the proposed *Maximum-Entropy* subsampling strategy employed by ADS against baseline strategies, keeping the generative model fixed to avoid confounding. Next, in Section 5.3, we evaluate the model end-to-end on both sampling and reconstruction by applying it in the real-world task of MRI acceleration with the fastMRI dataset, and comparing it to existing supervised approaches.

## 5.1 MNIST

In order to evaluate the effectiveness of the Maximum Entropy subsampling strategy employed by ADS, we compare it to two baseline subsampling strategies on the task of reconstructing images of digits from the MNIST dataset (LeCun et al., 1998). To this end, a diffusion model was trained on the MNIST training dataset, resized to $32 \times 32$ pixels. See Appendix A.2.1 for further details on training and architecture. Using this trained diffusion model, each subsampling strategy was used to reconstruct 500 unseen samples from the MNIST test set for various subsampling rates. Both pixel-based and line-based subsampling were evaluated, where line-based subsampling selects single-pixel-wide columns. The measurement model is thus $\mathbf{y}_t = \mathbf{U}(A_t)\mathbf{x}$, as there is no measurement noise or measurement transformation, i.e. $f(\mathbf{x}) = \mathbf{x}$. The baseline subsampling strategies used for comparison were as follows: (i) **Random** subsampling selects measurement locations from a uniform categorical distribution without replacement, and (ii) **Data Variance** subsampling selects measurement locations without replacement from a categorical distribution in which the probability of a given location is proportional to the variance across that location in the training set. This can therefore be seen as a data-driven but fixed design strategy.

Inference was performed using Diffusion Posterior Sampling for measurement guidance, with guidance weight $\zeta = 1$ and $T = 1000$ reverse diffusion steps. For ADS, measurements were taken at regular intervals in the window $[0, 800]$, with $N_p = 16$ particles and $\sigma_{\mathbf{y}} = 10$. For the fixed sampling strategies, the subsampling masks were set *a priori*, such that all diffusion steps are guided by the measurements, as is typical in inverse problem solving with diffusion models. The results to this comparison are illustrated by Figure 4, with numbers provided in Appendix A.3. We use Mean Absolute Error as an evaluation metric since MNIST consists of single channel brightness values. It is clear from these results that ADS outperforms fixed mask baselines, most notably in comparison with data-variance sampling: for pixel-based sampling, we find that maximum entropy sampling with a budget of 100 pixels outperforms data variance sampling with a budget of 250 pixels, i.e. actively sampling the measurements is as good as having $2.5\times$ the number of measurements with the data-variance strategy. We also find that the standard deviation of the reconstruction errors over the test set is significantly lower for 25% and 50% subsampling rates (typically $\sim$ 2-3$\times$ than baselines, leading to more reliable reconstructions). Further experiments on MNIST are carried out in Appendix A.3, where ADS is compared to *Active Deep Probabilistic Subsampling* Van Gorp et al. (2021), an end-to-end supervised method for active subsampling.

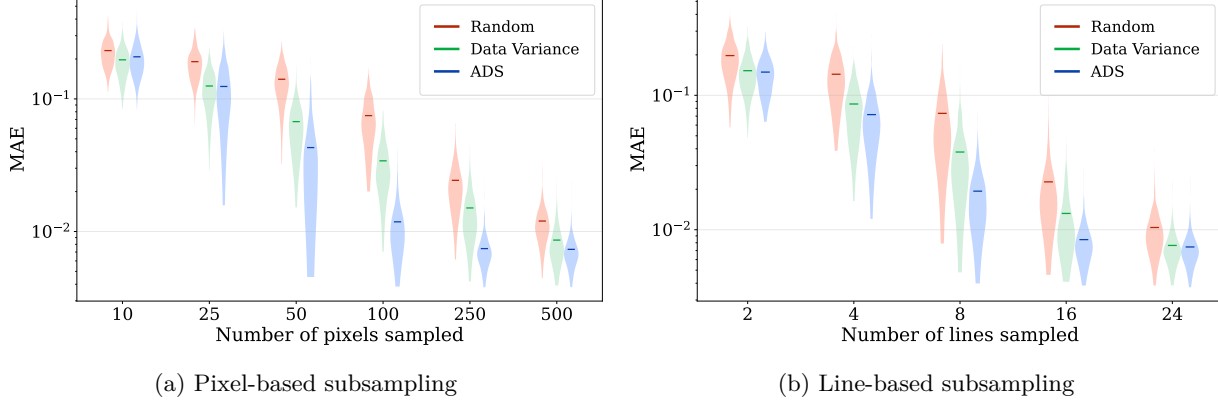

(a) Pixel-based subsampling

(b) Line-based subsampling

Figure 4: Comparison of ADS (ours) with two non-adaptive baselines. Evaluated based on reconstruction Mean Absolute Error (MAE) on $N = 500$ unseen samples from the MNIST test set. Note that MAE is plotted on a log scale.

## 5.2 CelebA

In order to evaluate ADS on a natural image dataset, a diffusion model was trained on the CelebA (Liu et al., 2015) training dataset at $128 \times 128$ resolution (see Appendix A.2.2 for training details). ADS was then benchmarked on N=100 samples from the CelebA test set against DPS with baseline sampling strategies.

This evaluation was carried out for a number sampling rates $|S| \in \{50, 100, 200, 300, 500\}$. The measurement scheme employed here samples 'boxes' of size 4×4 pixels from the image. The number of diffusion steps taken during inference was chosen based on sampling rate, with 400 steps for $|S| < 300$, 600 steps for $|S| = 500$. In each case, for sampling rate $|S|$ the sampling window $[10, |S| + 10]$ was used. These sampling windows were chosen empirically, finding that some initial unguided reverse diffusion steps help create a better initial estimate of the posterior. $N_p = 16$ particles with $\sigma_{\mathbf{y}} = 1$ were used to model the posterior distribution. The peak signal-to-noise ratio (PSNR) values between the posterior mean and test samples are provided in Figure 5a (a) and Table 1, with further metrics, including metrics across individual posterior samples, are provided in Appendix A.5. It is clear in these results that ADS outperforms baseline sampling strategies with DPS across a number of sampling rates. Note, for example, that ADS with 200 measurements achieves a higher PSNR than random-mask DPS with 300 measurements. In Figure 5b a selection of samples are provided to exemplify the mask designs produced by ADS. Note here that the masks focus on important features such as facial features in order to minimize the posterior entropy, leading to higher information gain about the target, and ultimately better recovery.

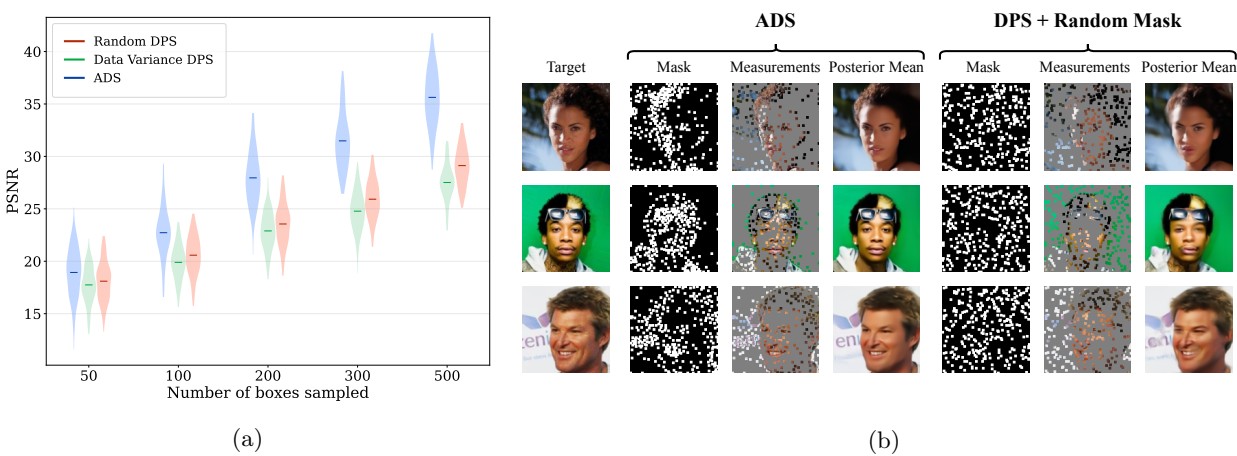

Figure 5: In (a), PSNR (↑) scores for ADS vs DPS with random and data variance sampling on N=100 unseen samples from the CelebA dataset are plotted, for increasing numbers of measurements. A measurement here is a $4 \times 4$ box of pixels. (b) shows some examples of ADS inference versus DPS with random measurements on the CelebA dataset from the evaluation with 200 boxes sampled.

| # Samples (Max %) | Random DPS | Data Variance DPS | ADS |
|---|---|---|---|
| **50 (4.88%)** | 18.099 (0.019) | 17.747 (0.018) | **18.932** (0.024) |
| **100 (9.76%)** | 20.580 (0.019) | 19.892 (0.017) | **22.725** (0.024) |
| **200 (19.53%)** | 23.555 (0.020) | 22.895 (0.018) | **27.954** (0.026) |
| **300 (29.29%)** | 25.919 (0.019) | 24.779 (0.019) | **31.483** (0.027) |
| **500 (48.82%)** | 29.123 (0.018) | 27.512 (0.018) | **35.630** (0.027) |

Table 1: PSNR (↑) on test samples from the CelebA dataset. Next to each # Samples $|S|$ is a 'Max %' indicating the maximum % of a $128 \times 128$ image that $|S|$ $4 \times 4$ boxes could cover.

## 5.3 MRI Acceleration

To assess the real-world practicability of ADS, it was evaluated on the popular fastMRI (Zbontar et al., 2018) $4\times$ acceleration benchmark for knee MRIs. In this task, one must reconstruct a fully-sampled knee MRI image given a budget of only 25% of the $\kappa$-space measurements, where each $\kappa$-space measurement is a vertical line of width 1 pixel. We compare with existing MRI acceleration methods focused specifically on learning sampling strategies, namely PG-MRI (Bakker et al., 2020), LOUPE (Bahadir et al., 2020), and SeqMRI (Yin et al., 2021), each of which are detailed in Appendix A.9. We use the same data train / validation

/ test split and data preprocessing as Yin et al. (2021) for comparability. In particular, the data samples are $\kappa$-space slices cropped and centered at $128 \times 128$, with $34,732$ train samples, $1,785$ validation samples, and $1,851$ test samples. We train a diffusion model on complex-valued image space samples $\mathbf{x} \in \mathbb{C}^{128 \times 128}$ obtained by computing the inverse Fourier transform of the $\kappa$-space training samples (see Appendix A.2.3 for further training details). The data space is therefore the complex image space, with $\kappa$-space acting as the measurement space. This yields the measurement model $\mathbf{y}_t = \mathbf{U}(A_t)\mathcal{F}(\mathbf{x}) + \mathbf{n}_t$, where $\mathcal{F}$ is the Discrete Fourier Transform, $\mathbf{n}_t \sim \mathcal{N}_{\mathcal{C}}(0, \sigma_{\mathbf{n}})$ is complex Gaussian measurement noise, and $\mathbf{U}(A_t)$ is the subsampling matrix selecting samples at indices $\in A_t$. ADS proceeds by running Diffusion Posterior Sampling in the complex image domain with guidance from $\kappa$-space measurements through the measurement model, selecting maximum entropy lines in the $\kappa$-space. We observed on data from the validation set that ADS reconstruction performance increases with the number of reverse diffusion steps, although with diminishing returns as steps increased. This indicates that in applying ADS, one can choose to increase sample quality at the cost of inference time and compute. To showcase the potential for ADS, we chose a large number of steps, $T = 10k$. Further, we choose guidance weight $\zeta = 0.85$, and sampling schedule $S$ evenly partitioning $[50, 2500]$, and an initial action set $A_{\text{init}} = \{63\}$, starting with a single central $\kappa$-space line. $N_p = 16$ and $\sigma_{\mathbf{y}} = 50$ were used to model the posterior distribution. Reconstructions were evaluated using the structural similarity index measure (SSIM) (Wang et al., 2004) to compare the absolute values of the fully-sampled target image and reconstructed image. The SSIM uses a window size of $4 \times 4$ with $k_1 = 0.01$ and $k_2 = 0.03$ as set be the fastMRI challenge. Table 2 shows the SSIM results on the test set, comparing ADS to recent supervised methods, along with a fixed-mask Diffusion Posterior Sampling using the same inference parameters to serve as a strong unsupervised baseline. The fixed-mask used with DPS measures the 8% of lines at the center of the $\kappa$-space, and random lines elsewhere, as used by Zbontar et al. (2018). It is clear from the results that ADS performs competitively with supervised baselines, and outperforms the fixed-mask diffusion-based approach. Figure 6 shows two reconstructions created by ADS. See Appendix A.11 for a histogram of all SSIMs over the test set for the diffusion-based approaches. Finally, we note the inference time for this model, which is an important factor in making active sampling worthwhile. Our model for fastMRI (Table 2) uses $\sim 40$ms / step with 76 steps per acquisition, leading to 3040ms per acquisition on our NVIDIA GeForce RTX 2080 Ti GPU. A typical acquisition time for a k-space line in MRI is 500ms-2500ms, or higher, depending on the desired quality (Jung & Weigel, 2013). Given modern hardware with increased FLOPs, we believe that this method is already near real-time, even without employing additional tricks to accelerate inference, such as quantization (Shang et al., 2023) or distillation (Salimans & Ho, 2022).

| Unsupervised | Method | SSIM ($\uparrow$) |
|:---:|:---:|:---:|
| ✗ | PG-MRI (Bakker et al., 2020) | 87.97 |
| ✗ | LOUPE (Bahadir et al., 2020) | 89.52 |
| ✓ | Fixed-mask DPS | 90.13 |
| ✗ | SeqMRI (Yin et al., 2021) | 91.08 |
| ✓ | ADS (Ours) | 91.26 |

Table 2: SSIM scores for fastMRI knee test set with 4x acceleration. See Appendix A.11 for SSIM histograms for diffusion-based methods.

## 5.4 Focused Ultrasound Scan-Line Subsampling

As a final experiment, we explore focused ultrasound scan-line subsampling (Huijben et al., 2020). In focused ultrasound, a set of focused scan-lines spanning a region of tissue is typically acquired, with the lines being displayed side-by-side to create an image. By acquiring only a small subset of these scan-lines, the power consumption and data transfer rate requirements of the probe can be reduced. Successfully applying scan-line subsampling could find applications in (i) extending battery life in wireless ultrasound probes (Baribeau et al., 2020), (ii) reducing bandwidth requirements for cloud-based ultrasound (Federici et al., 2023), and (iii) increasing frame-rates in 3D ultrasound (Giangrossi et al., 2022). In this experiment, we demonstrate using ADS to identify informative subsets of scan-lines, from which the full image can be reconstructed.

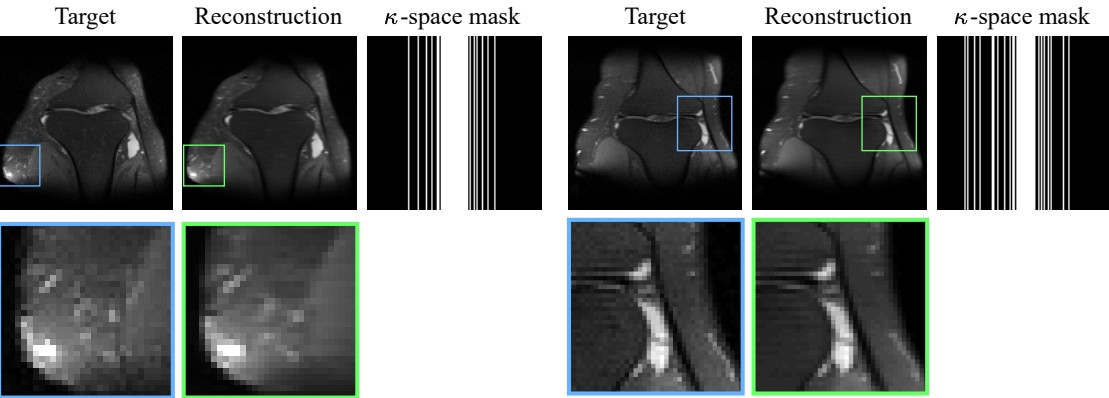

Figure 6: Sample fastMRI reconstructions produced by ADS, including the generated $\kappa$-space masks. The SSIMs are 95.8 for the left, and 94.4 for the right.

To this end, we trained a diffusion model on the samples from the CAMUS (Leclerc et al., 2019) dataset, with the train set consisting of 7448 frames from cardiac ultrasound scans across 500 patients, resized to $128 \times 128$ in the polar domain. Each column of pixels in the polar domain is taken to represent a single beamformed scan-line, yielding the model $\mathbf{y}_t = \mathbf{U}(A_t)\mathbf{x}$, where $\mathbf{x}$ is the fully-sampled target image, $\mathbf{U}(A_t)$ is a mask selecting a set of scan-lines, and $\mathbf{y}_t$ is the set of scan-line measurements selected by $A_t$. Inference takes place in the polar domain, and results are scan-converted for visualization. For ADS inference, the parameters chosen were $T = 400$ steps, $S$ evenly partitioning the interval $[0, 320]$, $N_p = 16$, $\sigma_y = 1$, and $\zeta = 10$. We benchmark ADS against DPS using random and data-variance fixed-mask strategies on a test set consisting of frames from 50 unseen patients, finding that ADS significantly outperforms both in terms of reconstruction quality metrics. Figure 7 presents the PSNR results along with a sample reconstruction produced by ADS versus random fixed-mask DPS, with further quantitative results and sample outputs provided in Appendix A.5.

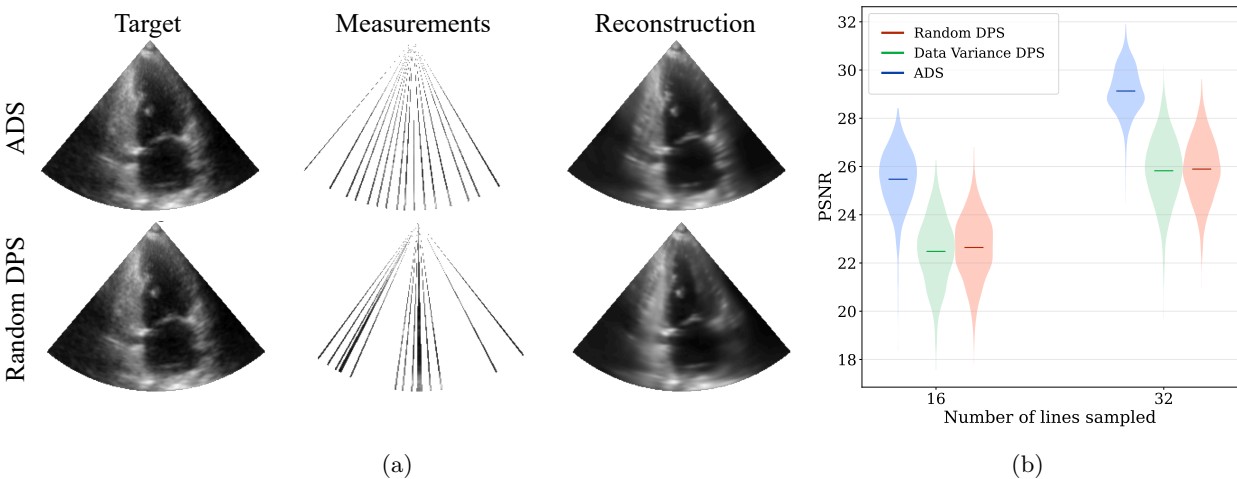

Figure 7: In (a), a comparison of selected measurements and reconstructions generated by ADS versus DPS with a random mask on the CAMUS (Leclerc et al., 2019) dataset. The measurement budget is $16/128 = 12.5\%$ of the scan-lines. (b) shows the PSNR scores across a held-out test set consisting of 875 frames from 50 unseen patients, with measurement budgets of 16 and 32 scan lines, representing subsampling rates of 12.5% and 25%, respectively.

## 6 Discussion

While ADS appears to outperform fixed-mask baselines on MNIST, CelebA, and fastMRI, it is interesting to note that the relative improvement offered by ADS is less pronounced in the case of fastMRI. For line-based image subsampling on MNIST with 25% samples, ADS achieves a 50% reduction in reconstruction error versus a fixed-mask approach (MAE = 0.019 vs 0.037), whereas with line-based $\kappa$-space subsampling for fastMRI with 25% samples, ADS achieves only a 12% relative improvement (SSIM = 91.26 vs 90.13). We find that the size of this performance gap between fixed and active mask design strategies can be explained by examining the distribution of masks designed by ADS on each task (See Appendix A.10). Indeed, the masks designed for fastMRI are very similar, whereas those designed for MNIST typically differ depending on the sample. When mask designs are similar, then fixed masks will perform similarly to actively designed masks. This is in part a feature of the data distribution – for example, most information in $\kappa$-space is contained in the center, at lower frequencies. Tasks in which one might expect significantly better performance from ADS are therefore those in which optimal masks will be highly sample-dependent.

Another interesting trend appears in the results when observing the relative improvement of ADS over fixed-mask strategies as a function of the number of samples taken. One might expect that the relative improvement is highest when the subsampling rate is lowest, monotonically decreasing as more samples are taken. In fact, however, we observe that the largest relative improvement appears not at the lowest subsampling rates, but rather at medium rates, e.g. 10% - 50%. One possible explanation for this is the under-performance of diffusion posterior sampling in scenarios with very few measurements, leading to inaccurate estimates of the posterior distribution and therefore sub-optimal sampling strategies. Further study of the performance of posterior sampling methods in cases with very few measurements could provide an interesting direction for future work.

We also observe that data variance sampling outperforms random sampling on MNIST, but not on CelebA. This somewhat surprising result can be explained by noting that the variance in MNIST appears in a highly informative region, namely around the center of the images where the digits are contained. In contrast, the variance in CelebA appears mostly in the background, where backgrounds may exhibit large deviations in brightness and color. Despite the large variance in the backgrounds, they are typically homogeneous blocks of color that can be inpainted well from sparse samples. Hence, taking most samples in this high-variance background region leading to suboptimal sampling strategies and reconstructions.

A final point of discussion is on choosing the number of particles $N_p$ for a particular task. We find in Appendix A.5 that 8-16 particles are sufficient for modelling the posterior in CelebA, with diminishing returns for increasing $N_p$. In general, we advise choosing $N_p$ such that the modes of the posterior distribution can be sufficiently represented, so that regions of disagreement between modes can be identified by the entropy computation. For example, an MNIST dataset containing only 2 digits will require fewer particles than an MNIST dataset containing 10.

In conclusion, we have proposed a method for using diffusion models as active subsampling agents without the need for additional training, using a simple, interpretable action selection policy. We show that this method significantly outperforms fixed-mask baselines on MNIST, CelebA, and ultrasound scan-line subsampling, and competes with existing supervised approaches in MRI acceleration without tasks-specific training. This method therefore takes a step towards transparent active sensing with automatically generated adaptive strategies, decreasing cost factors such as exposure time, and energy usage.

## 7 Limitations & Future Work

While experiments in Section 5 evidence some strengths of ADS against baseline sampling strategies, it is not without limitations. For example, the duration of inference in ADS is dependent on that of the diffusion posterior sampling method. Since low latency is essential in active subsampling, future work could aim to accelerate posterior sampling with diffusion models, leading to accelerated ADS. Another limitation is that the number of measurements taken is upper-bounded by the number of reverse diffusion steps $T$; this limitation could be overcome by extending ADS to generate *batch designs* (Azimi et al., 2010), containing multiple measurements, from a single posterior estimate. Future work applying ADS in diverse domains

would also help to further assess the robustness of the method. Finally, as mentioned in Section 6, we note that ADS does not show significant improvements over baseline sampling strategies under very low sampling rates, e.g. $< 5\%$. We hypothesize that this may be due to inaccurate posterior sampling with very few measurements, and believe that this presents an interesting direction for future study.

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

# A  Appendix

## A.1  Derivation of Equation (10)

Here we show that maximising the policy function does not require computing a set of particles for each possible action in the case where the action is a subsampling mask. Because the subsampling mask $A_t = A_{t-1} \cup a_t$ only varies in $a_t$ in the arg max, the elements of each particle $\hat{\mathbf{y}}^{(i)}$ will remain the same for each possible $A_t$ except for at those indices selected by $a_t$. We therefore decompose the squared L2 norm into two squared L2 norms, one for the indices in $a_t$ and the other for those in $A_{t-1}$. The latter then becomes a constant in the argmax, and can be ignored. This results in a formulation in which we only need to compute the squared L2 norms for the set of elements corresponding with $a_t$. We use $\mathbf{U}(A_t)$ to indicate the subsampling matrix containing 1s on the diagonal at indices in $A_t$.

$$
\begin{aligned}
a_t^* &= \arg\max_{a_t} \sum_i^{N_p} \log \sum_j^{N_p} \exp\left\{ \frac{||\hat{\mathbf{y}}_t^{(i)} - \hat{\mathbf{y}}_t^{(j)}||_2^2}{2\sigma_y^2} \right\} \\
&= \arg\max_{a_t} \sum_i^{N_p} \log \sum_j^{N_p} \exp\left\{ \frac{||\mathbf{U}(A_t)f(\hat{\mathbf{x}}_0^{(i)}) - \mathbf{U}(A_t)f(\hat{\mathbf{x}}_0^{(j)})||_2^2}{2\sigma_y^2} \right\} \\
&= \arg\max_{a_t} \sum_i^{N_p} \log \sum_j^{N_p} \exp\left\{ \frac{\sum_{k \in A_t} (f(\hat{\mathbf{x}}_0^{(i)})_k - f(\hat{\mathbf{x}}_0^{(j)})_k)^2}{2\sigma_y^2} \right\} \\
&= \arg\max_{a_t} \sum_i^{N_p} \log \sum_j^{N_p} \exp\left\{ \frac{\sum_{l \in a_t} (f(\hat{\mathbf{x}}_0^{(i)})_l - f(\hat{\mathbf{x}}_0^{(j)})_l)^2 + \sum_{m \in A_{t-1}} (f(\hat{\mathbf{x}}_0^{(i)})_m - f(\hat{\mathbf{x}}_0^{(j)})_m)^2}{2\sigma_y^2} \right\} \\
&= \arg\max_{a_t} \sum_i^{N_p} \log \sum_j^{N_p} \left( \exp\left\{ \frac{\sum_{l \in a_t} (f(\hat{\mathbf{x}}_0^{(i)})_l - f(\hat{\mathbf{x}}_0^{(j)})_l)^2}{2\sigma_y^2} \right\} \exp\left\{ \frac{\sum_{m \in A_{t-1}} (f(\hat{\mathbf{x}}_0^{(i)})_m - f(\hat{\mathbf{x}}_0^{(j)})_m)^2}{2\sigma_y^2} \right\} \right) \\
&= \arg\max_{a_t} \sum_i^{N_p} \log \sum_j^{N_p} \exp\left\{ \frac{\sum_{l \in a_t} (f(\hat{\mathbf{x}}_0^{(i)})_l - f(\hat{\mathbf{x}}_0^{(j)})_l)^2}{2\sigma_y^2} \right\} \\
&= \arg\max_{a_t} \sum_i^{N_p} \log \sum_j^{N_p} \exp\left\{ \frac{\sum_{l \in a_t} (\hat{\mathbf{y}}_l^{(i)} - \hat{\mathbf{y}}_l^{(j)})^2}{2\sigma_y^2} \right\}
\end{aligned}
\tag{11}
$$

## A.2  Training Details

The methods and models are implemented in the Keras 3.1 (Chollet et al., 2015) library using the Jax backend (Bradbury et al., 2018). The DDIM architecture is provided by Keras3 at the following URL: `https://keras.io/examples/generative/ddim/`. Each model was trained using one GeForce RTX 2080 Ti (NVIDIA, Santa Clara, CA, USA) with 11 GB of VRAM.

### A.2.1  MNIST

The model was trained for 500 epochs with the following parameters: widths=[32, 64, 128], block_depth=2, diffusion_steps=30, ema_0.999, learning_rate=0.0001, weight_decay=0.0001, loss="mae".

### A.2.2 CelebA

The model was trained for 200 epochs with the following parameters: widths=[32, 64, 96, 128], block_depth=2, diffusion_steps=30, ema_0.999, learning_rate=0.0001, weight_decay=0.0001, loss="mae".

### A.2.3 FastMRI

The training run of 305 epochs with the following parameters: widths=[32, 64, 96, 128], block_depth=2, diffusion_steps=30, ema_0.999, learning_rate=0.0001, weight_decay=0.0001, loss="mae".

### A.3 MNIST metrics

| # Pixels (%) | Random | Data Variance | ADS (Ours) |
|---|---|---|---|
| 10 (.97%) | 0.231 (.002) | **0.197** (.002) | 0.207 (.002) |
| 25 (2.44%) | 0.190 (.002) | 0.125 (.002) | **0.124** (.002) |
| 50 (4.88%) | 0.140 (.002) | 0.067 (.001) | **0.042** (.001) |
| 100 (9.76%) | 0.074 (.001) | 0.034 (.001) | **0.011** (.000) |
| 250 (24.41%) | 0.024 (.000) | 0.015 (.000) | **0.007** (.000) |
| 500 (48.82%) | 0.011 (.000) | 0.008 (.000) | **0.007** (.000) |
| **# Lines (%)** | **Random** | **Data Variance** | **ADS (Ours)** |
| 2 (6.25%) | 0.197 (.003) | 0.152 (.002) | **0.148** (.002) |
| 4 (12.5%) | 0.143 (.002) | 0.086 (.002) | **0.071** (.001) |
| 8 (25%) | 0.073 (.001) | 0.037 (.001) | **0.001** (.000) |
| 16 (50%) | 0.022 (.001) | 0.013 (.000) | **0.000** (.000) |
| 24 (75%) | 0.010 (.000) | 0.0076 (.000) | **0.0074** (.000) |

Table 3: Mean and (standard error) for the Mean Absolute Error ($\downarrow$) in reconstruction of MNIST samples, using pixel- and line-based subsampling.

### A.4 Comparison with *Active Deep Probabilistic Subsampling*

In this appendix we provide qualitative and quantitative results comparing ADS to a supervised active sampling approach, *Active Deep Probabilistic Subsampling* (ADPS) by Van Gorp et al. (2021). ADPS performs active sampling by training a Long Short-Term Memory (LSTM) model to produce logits for each sampling location at each iteration. This is trained jointly with a 'task model', which performs some task based on the partial observations. This could be, for example, a reconstruction model. We chose to compare ADS to ADPS on the task of reconstructing MNIST digits to facilitate a qualitative comparison of the masks generated by each method. In order to compare on a reconstruction task, we adapted the MNIST classification model provided in the ADPS codebase[1] for reconstruction by replacing their fully-connected classification layers with a large UNet and using an MSE training objective to reconstruct $32\times32$ pixel MNIST digits from partial observations containing $81/1024 \approx 8\%$ of the pixels. We chose a UNet architecture with widths=[32, 64, 128, 256] and block_depth=2 as the reconstruction model for ADPS, making it similar to the UNet denoising network used by ADS. Both models were trained on the MNIST training set and evaluated on 1000 unseen samples from the validation set.

The results are presented in Figures 8, 9, 10, and 11, indicating that ADS outperforms ADPS in terms of reconstruction error, and generated more adaptive and intuitive masks. One reason for the limited adaptivity in the masks generated by ADPS could be the fact that the reconstruction model is trained jointly with the sampling model, leading to a strategy wherein the reconstruction model can depend on receiving certain inputs and not others. We note, however, that the performance of ADPS could change depending on the choice of reconstruction model, loss function used, and other design choices. In general, the major advantage

---

[1]https://github.com/IamHuijben/Deep-Probabilistic-Subsampling/tree/master/ADPS_vanGorp2021/MNIST_Classification

of ADS over ADPS is that it does not need to be re-trained for different choices of forward model or subsampling rates: all that's needed is a diffusion prior, and sampling schemes can vary freely at inference time.

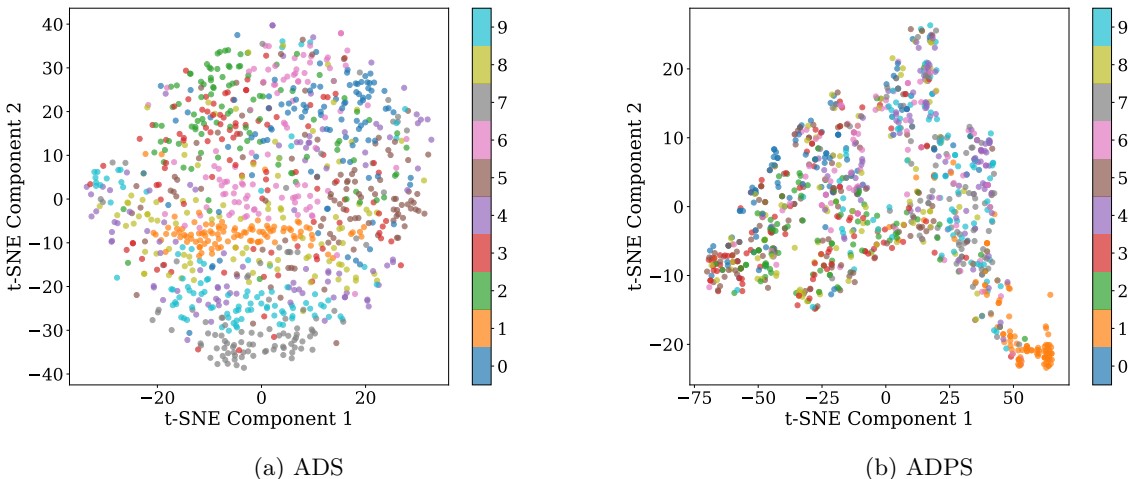

(a) ADS

(b) ADPS

Figure 8: t-SNE plots for the masks generated by both (a) ADS and (b) ADPS for 1000 unseen samples from the MNIST validation set. The t-SNE plot for ADS appears to exhibit stronger clustering between different digits, indicating that the generated masks are more digit-dependent. ADPS creates a distinct mask for the digit '1', but there is much overlap across other digits.

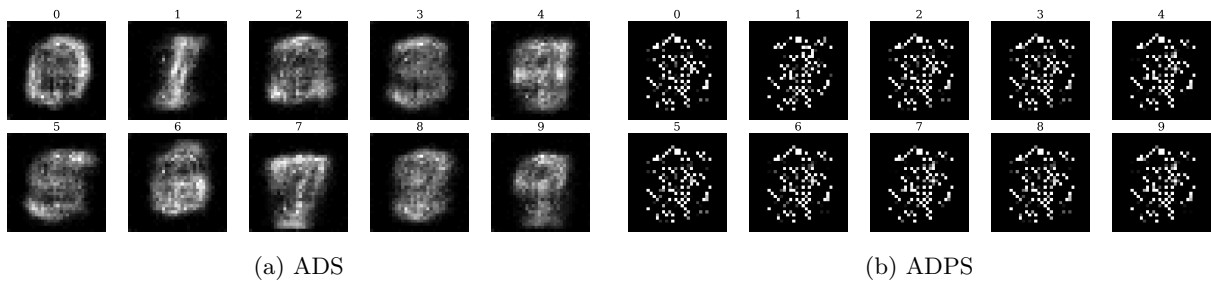

(a) ADS

(b) ADPS

Figure 9: The mean across all masks for each digit generated by (a) ADS and (b) ADPS. ADS has generated masks that trace the digits, whereas ADPS has generated spread-out masks in the central region of the image where digits may appear.

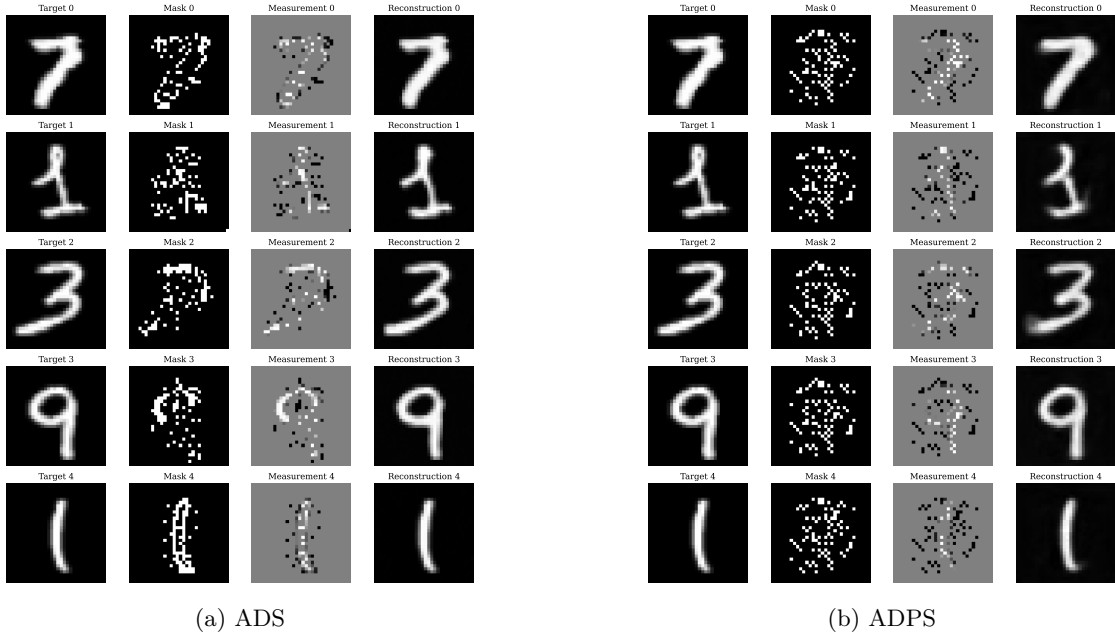

(a) ADS

(b) ADPS

Figure 10: Random samples from the test set with masks, measurements, and reconstructions generated by (a) ADS and (b) ADPS.

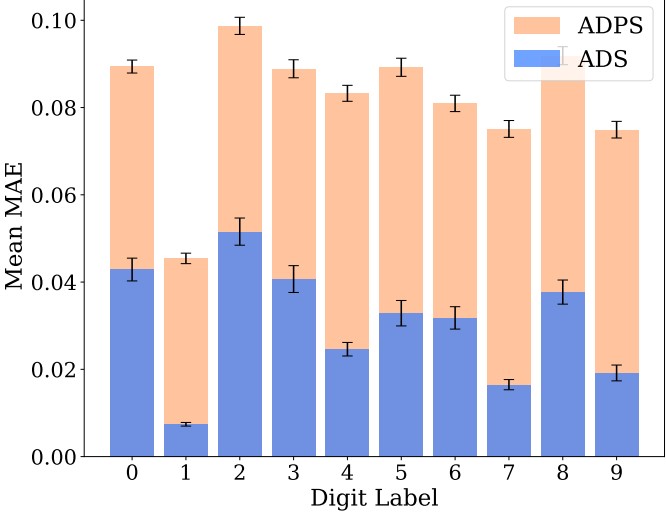

Figure 11: Digit-wise comparison of Mean Absolute Error between targets and reconstructions generated by ADS versus ADPS. The error bars on each bar indicate the standard error of the mean.

### A.5 CelebA Further Metrics

Table 4: MAE (↓) of the posterior mean on targets from the CelebA test set.

| Num Samples | Random DPS | Data Variance DPS | ADS |
|---|---|---|---|
| **50** | 19.768 (0.043) | 20.625 (0.044) | 19.569 (0.060) |
| **100** | 13.623 (0.028) | 14.809 (0.028) | 11.731 (0.034) |
| **200** | 8.909 (0.020) | 9.786 (0.020) | 6.356 (0.018) |
| **300** | 6.538 (0.014) | 7.507 (0.016) | 4.426 (0.012) |
| **500** | 4.320 (0.009) | 5.185 (0.010) | 2.825 (0.007) |

Table 5: SSIM (↑) of the posterior mean on targets from the CelebA test set.

| Num Samples | Random DPS | Data Variance DPS | ADS |
|---|---|---|---|
| **50** | 0.577 (0.001) | 0.557 (0.001) | 0.567 (0.001) |
| **100** | 0.687 (0.001) | 0.652 (0.001) | 0.697 (0.001) |
| **200** | 0.791 (0.001) | 0.759 (0.000) | 0.831 (0.001) |
| **300** | 0.851 (0.000) | 0.820 (0.000) | 0.889 (0.000) |
| **500** | 0.911 (0.000) | 0.881 (0.000) | 0.941 (0.000) |

Table 6: LPIPS (↓) (Zhang et al., 2018) of the posterior mean on targets from the CelebA test set.

| Num Samples | Random DPS | Data Variance DPS | ADS |
|---|---|---|---|
| **50** | 0.306 (0.001) | 0.320 (0.000) | 0.287 (0.001) |
| **100** | 0.230 (0.000) | 0.248 (0.000) | 0.190 (0.001) |
| **200** | 0.152 (0.000) | 0.174 (0.000) | 0.103 (0.000) |
| **300** | 0.111 (0.000) | 0.131 (0.000) | 0.071 (0.000) |
| **500** | 0.069 (0.000) | 0.086 (0.000) | 0.043 (0.000) |

Table 7: Mean MAE (↓) across posterior samples on targets from the CelebA test set.

| Num Samples | Random DPS | Data Variance DPS | ADS |
|---|---|---|---|
| **50** | 24.486 (0.048) | 25.321 (0.050) | 23.895 (0.067) |
| **100** | 17.436 (0.035) | 18.493 (0.034) | 14.463 (0.040) |
| **200** | 11.255 (0.025) | 12.337 (0.024) | 7.704 (0.021) |
| **300** | 8.230 (0.018) | 9.372 (0.020) | 5.225 (0.014) |
| **500** | 5.389 (0.011) | 6.477 (0.013) | 3.268 (0.008) |

Table 8: Mean PSNR (↑) across posterior samples on targets from the CelebA test set.

| Num Samples | Random DPS | Data Variance DPS | ADS |
|---|---|---|---|
| **50** | 15.988 (0.016) | 15.752 (0.015) | 16.971 (0.022) |
| **100** | 18.169 (0.017) | 17.791 (0.016) | 20.828 (0.023) |
| **200** | 21.237 (0.019) | 20.621 (0.016) | 26.314 (0.025) |
| **300** | 23.617 (0.018) | 22.604 (0.018) | 30.160 (0.026) |
| **500** | 26.795 (0.018) | 25.274 (0.018) | 34.546 (0.026) |

Table 9: Mean SSIM (↑) across posterior samples on targets from the CelebA test set.

| Num Samples | Random DPS | Data Variance DPS | ADS |
|---|---|---|---|
| **50** | 0.486 (0.001) | 0.471 (0.001) | 0.481 (0.001) |
| **100** | 0.599 (0.001) | 0.569 (0.001) | 0.623 (0.001) |
| **200** | 0.725 (0.001) | 0.691 (0.001) | 0.783 (0.001) |
| **300** | 0.799 (0.000) | 0.764 (0.000) | 0.859 (0.001) |
| **500** | 0.877 (0.000) | 0.841 (0.000) | 0.925 (0.000) |

Table 10: Mean LPIPS (↓) (Zhang et al., 2018) across posterior samples on targets from the CelebA test set.

| Num Samples | Random DPS | Data Variance DPS | ADS |
|---|---|---|---|
| **50** | 0.321 (0.001) | 0.329 (0.001) | 0.305 (0.001) |
| **100** | 0.250 (0.000) | 0.260 (0.000) | 0.208 (0.001) |
| **200** | 0.172 (0.000) | 0.184 (0.000) | 0.112 (0.000) |
| **300** | 0.127 (0.000) | 0.141 (0.000) | 0.073 (0.000) |
| **500** | 0.079 (0.000) | 0.095 (0.000) | 0.041 (0.000) |

## A.6 Further Results on Ultrasound Scan-Line Subsampling

Table 11: MAE on test samples from the CAMUS dataset.

| Num Samples | Random DPS | Data Variance DPS | ADS |
|---|---|---|---|
| **16** | 12.198 (0.002) | 12.431 (0.002) | 9.533 (0.001) |
| **32** | 8.520 (0.001) | 8.599 (0.001) | 6.701 (0.001) |

Table 12: PSNR on test samples from the CAMUS dataset.

| Num Samples | Random DPS | Data Variance DPS | ADS |
|---|---|---|---|
| **16** | 22.643 (0.002) | 22.480 (0.002) | 25.473 (0.001) |
| **32** | 25.893 (0.001) | 25.824 (0.002) | 29.130 (0.001) |

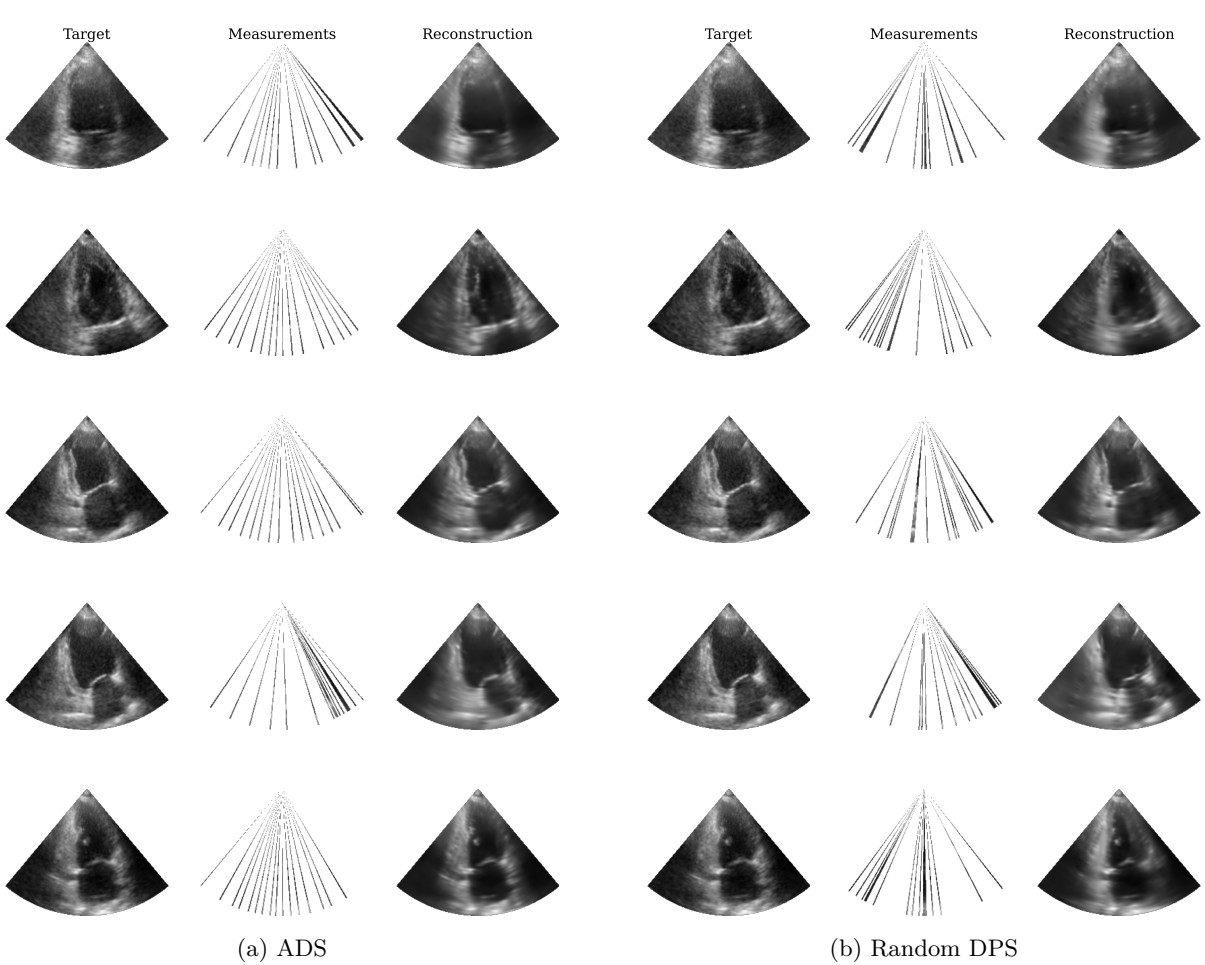

(a) ADS  (b) Random DPS

Figure 12: Targets, measurements, and reconstructions for 5 random samples from the CAMUS test set by (a) ADS and (b) Random fixed-mask DPS. The Mean Absolute Error for each reconstruction is provided.

## A.7 Runtime for CelebA

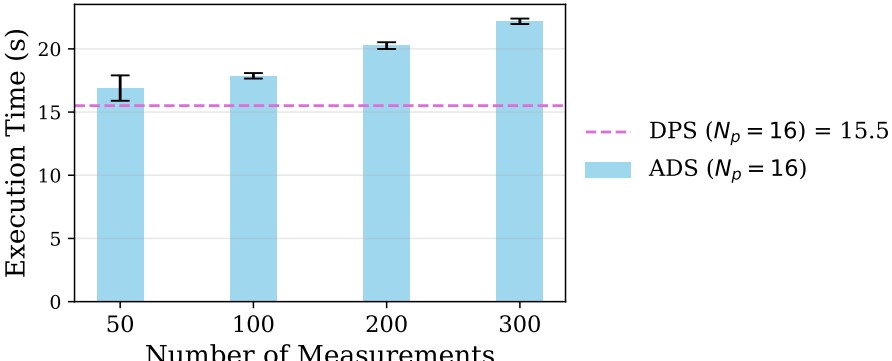

Figure 13: Runtime of ADS on samples from the CelebA dataset with size $128 \times 128$ pixels, using an NVIDIA GeForce RTX 2080 Ti GPU. Note the contribution in runtime from DPS, below the dashed purple line, and the contribution from ADS, above the dashed purple line, which scales linearly with the number of measurements. The execution time presented is the mean execution time over $N = 20$ runs, with 1 standard deviation indicated by the error bars.

## A.8 Effect of varying the number of particles

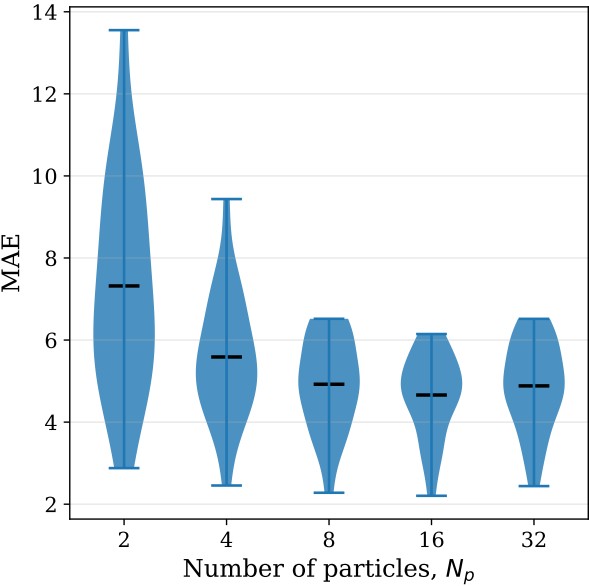

Figure 14: Reconstruction accuracy on $N = 20$ samples from the the CelebA validation set, as measured by the MAE between the target and posterior mean, using a variety of values for $N_p$.

### A.9 FastMRI Comparison Methods

### A.9.1 PG-MRI

Bakker et al. (2020) use policy-gradient methods from Reinforcement Learning to learn a policy function $\pi_\phi(a_t \mid \hat{x}_t)$ that outputs new measurement locations given the current reconstruction. Reconstructions are then generated using a pre-existing U-Net based reconstruction model provided by the fastMRI repository.

### A.9.2 LOUPE

LOUPE (Bahadir et al., 2020) introduces a end-to-end learning framework that trains a neural network to output under-sampling masks in combination with an anti-aliasing (reconstuction) model on undersampled full-resolution MRI scans. Their loss function consists of a reconstruction term and a trick to enable sampling a mask.

### A.9.3 SeqMRI

With SeqMRI, Yin et al. (2021) propose an end-to-end differentiable sequential sampling framework. They therefore jointly learn the sampling policy and reconstruction, such that the sampling policy can best fit with the strengths and weaknesses of the reconstruction model, and vice versa.

## A.10 ADS Mask Distributions for MNIST and fastMRI

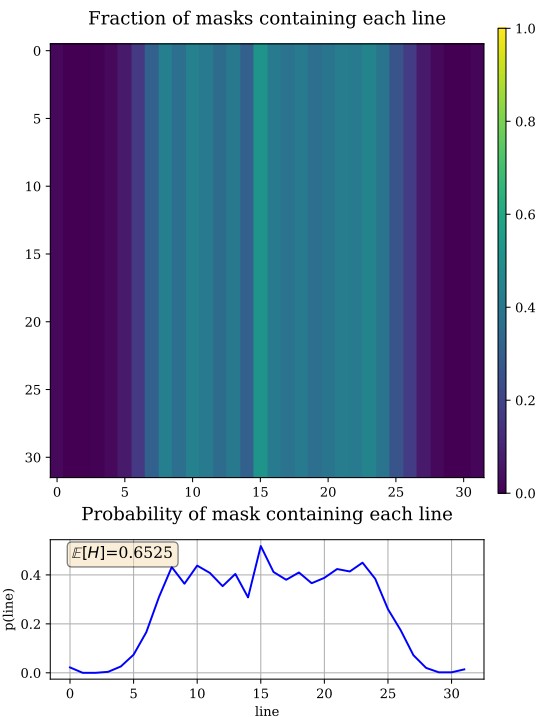

(a) ADS mask distribution for 500 samples from the MNIST test set.

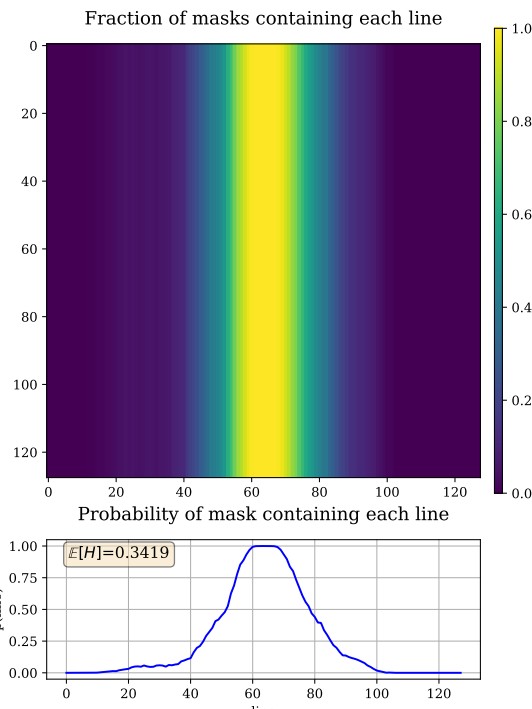

(b) ADS mask distribution for 1851 samples from the fastMRI test set.

Figure 15: The distribution of masks chosen by ADS varies according to the task. We observe that the masks chosen for MNIST are less predictable a priori than those chosen for fastMRI, leading to a stronger performance by ADS relative to fixed-mask approaches. We plot at the bottom of each plot estimates of the probability that each line will appear in a mask generated by ADS as Bernoulli variables (either the line is present in the mask, or not). To quantify the predictability of these masks, we compute the average entropy over each of these variables, finding that MNIST masks are signficantly less predictable than those for fastMRI.

## A.11   FastMRI SSIM Distributions

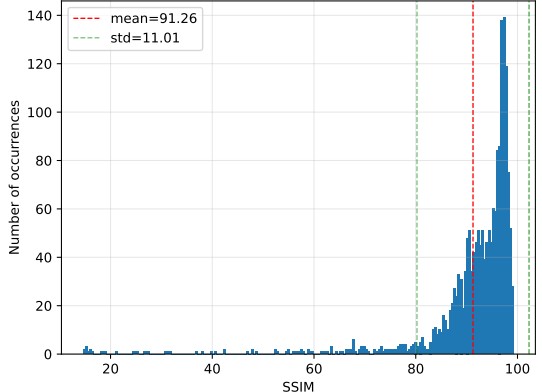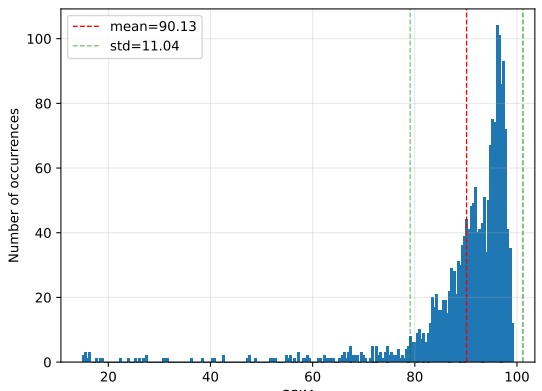

(a) 200-bin histogram showing distribution of SSIM scores across the FastMRI knee test set for 4x acceleration using ADS.

(b) 200-bin histogram showing distribution of SSIM scores across the FastMRI knee test set for 4x acceleration using diffusion posterior sampling with fixed $\kappa$-space masks.

Figure 16: Histograms comparing the distribution of SSIM scores using ADS and diffusion posterior sampling.

