# OpenReview forum: "Active Diffusion Subsampling"
_TMLR — Accepted by TMLR_

### Review · Reviewer_1Wyn · 2024-11-24

**Summary Of Contributions:**

This work proposes Active Diffusion Subsampling (ADS), which is an greedy adaptive scheme for subsampling measurements in diffusion posterior sampling, such as patches in images. To approximate the entropy and thus select patches that maximize information gain, the authors propose using a Gaussian mixture model to model the uncertainty when reconstructing with incomplete measurements. The experiments demonstrate significantly better reconstruction quality when given a limited sample budget, showing heuristically that it can find areas of interest, and can presumably be extended to work with pre-trained diffusion models.

**Audience:**

Yes

**Claims And Evidence:**

Yes

**Requested Changes:**

Overall, I am positive on this work, as it clearly demonstrates that it is possible to find areas of interest for diffusion posterior models. The only critical change I request is to clarify the exposition of the proposed ADS method, in particular the roles of all the different variables $N_p, S,y_\tau, \sigma_y$ etc.

**Strengthening**
* Additional experiments as suggested above, such as varying $N_p$, wall-clock comparison to fixed subsamplings, connection with previous work
* Fix the caption of Figure 3
* Clarifying the contributions (adaptively deriving a good subsampling regime for a given reconstruction task) in the abstract and introduction
* Add more comments into Algorithm 1 e.g. what is step 7 doing? step 8: subsampled data fidelity step, *single input* $\mathbf{x}$

**References**

[1] Efron, Bradley. "Tweedie’s formula and selection bias." Journal of the American Statistical Association 106.496 (2011): 1602-1614.

[2] Robbins, Herbert E. "An empirical Bayes approach to statistics." Breakthroughs in Statistics: Foundations and basic theory. New York, NY: Springer New York, 1992. 388-394.

[3] Van Gorp, H., Huijben, I., Veeling, B. S., Pezzotti, N., & Van Sloun, R. J. (2021, July). Active deep probabilistic subsampling. In International Conference on Machine Learning (pp. 10509-10518). PMLR.

**Strengths And Weaknesses:**

**Strengths**

* The work is generally well-presented with sufficient background. The corresponding motivation is clear and the derivation of the ADS method is intuitive from a high level. The work appears to be novel, extending previous subsampling work on end-to-end models and GANs to diffusion models.
* The experiments are convincing and clearly show that ADS is able to find informative patches of images for reconstruction, in particularly in limited budget scenarios.
* The training protocol and additional experimental detail look complete and should be reproducible.

**Weaknesses**

* While the abstract details ADS as a method for active subsampling, it mentions the term "active subsampling" exactly once, and does not mention the experimental contributions of the work (presumably acceleration during inference). Moreover, the use of the word "outperform" is not attached to any metric such as speed or error. Please clarify that ADS is able to greedily find subsamplings that are "more informative" as to have better reconstruction.

* (Sec. 4.1) The actual explanation of ADS is a bit unclear. I initially (wrongly) believed that ADS was trying to find an optimal set of subsampled timesteps $S$ such that reconstructed $\mathbf{x}_0$ would be of high quality. However upon further reading into the experiments section, it appears actually that $|S|$ controls the number of allowable subsamplings, and $N_p$ is a set of possible reconstructions, such as $N_p=4$ in Figure 1. Moreover, the GMM is directly stated to have variances $\sigma_y^2 I$ centered at the current "particles", but it is not clear what $\sigma_y$ actually is, perhaps it is connected to the current time-step $\tau \in S$? I would suggest rewriting Section 4.1 and clarifying the role of $S$ earlier with a sentence.

* Maybe I am still misunderstanding, but it appears that to compute the region of maximum uncertainty, fully-sampled measurements are still required (stated in page 6: "compute a single set of fully-sampled measurement particles ..."). Could the authors please clarify this, especially how a fully-sampled measurement fits into the whole pipeline?

* There is a lack of comparison with other learned subsampling methods, notably adaptive in [3]. While I appreciate that the policy is black-box, the end results seems to be a working subsampling mask for a given reconstruction model, and it would be very interesting to see how the derived masks work between different model archetypes.

**Minor**

* Figure 3 is quite hard to understand. Please add a textual description in the caption.

* I would be interested to see how much longer ADS takes compared to the baseline fixed subsampling due to the additional particles required.

* How does the double sum over the number of samples in Equation 10 (sum over $i,j=1,...,N_p$) fare in practice? It seems that in high dimensions, $N_p$ needs to be higher, which would lead to worse performance. Also, how does the method differ for different values of $N_p$?

**Questions/Comments**

* (Sec. 3.3) Above equation 5: "The (variance-preserving) SDE defining the noising process is as follows:..."

* Missing reference for Tweedie's formula on page 5, e.g. [1,2]. Maybe a fuller explanation specifying the relation between the MMSE denoiser and the score function would aid the flow.

* What is $N_p$ for the MRI experiment?

* Can the method be extended to finding common subsampling masks for multiple images? It would be interesting to see how the masks found by ADS differ and presumably degrade once a common mask for multiple images is deployed.

* There are some minor typos in the references, "{mri}", missing captialisation on names like "bayesian" etc.

---

> ### Author Response · Authors · 2025-01-23
> **Part 1**
>
> Thank you for your helpful suggestions and insights, which have contributed to the clarity and completeness of the paper. In the following, we address each of your comments.
>
> > *While the abstract details ADS as a method for active subsampling, it mentions the term "active subsampling" exactly once, and does not mention the experimental contributions of the work (presumably acceleration during inference). Moreover, the use of the word "outperform" is not attached to any metric such as speed or error. Please clarify that ADS is able to greedily find subsamplings that are "more informative" as to have better reconstruction.*
>
> We have now updated the abstract to clarify what is meant by 'active subsampling', explaining that it generates informative subsampling masks, and tied the claims about outperforming baselines to the specific image quality metrics used.
>
> > *(Sec. 4.1) The actual explanation of ADS is a bit unclear. I initially (wrongly) believed that ADS was trying to find an optimal set of subsampled timesteps S such that reconstructed x0 would be of high quality. However upon further reading into the experiments section, it appears actually that |S| controls the number of allowable subsamplings, and Np is a set of possible reconstructions, such as Np=4 in Figure 1. Moreover, the GMM is directly stated to have variances σy2I centered at the current "particles", but it is not clear what σy actually is, perhaps it is connected to the current time-step τ∈S? I would suggest rewriting Section 4.1 and clarifying the role of S earlier with a sentence.*
>
> We have re-written Section 4.1 to improve our explanation of ADS, now providing an illustrative example to show how the subsampling schedule S specifies the timesteps at which new measurements should be acquired. In addition, we have clarified that $\sigma_y$ is a hyper-parameter, and included an additional paragraph to provide intuition on how it affects the entropy computation (namely by controlling its sensitivity to outliers).
>
> > *Maybe I am still misunderstanding, but it appears that to compute the region of maximum uncertainty, fully-sampled measurements are still required (stated in page 6: "compute a single set of fully-sampled measurement particles ..."). Could the authors please clarify this, especially how a fully-sampled measurement fits into the whole pipeline?*
>
> This is an important point which we agree could be explained more clearly in the paper. These fully-sampled measurements used to compute regions of uncertainty are in fact *simulated* fully-sampled measurements, produced by passing the set of particles / posterior samples $\{x^{(i)}\}$ through the forward model. Because the forward model is linear Gaussian, we know that sampling the region where we are most uncertain about the outcome of a measurement will result in maximum information gain.  We have now made this clearer by (i) adding a new comment to the algorithm highlighting where this measurement simulation is performed, and (ii) added new labels and a detailed caption to Figure 3, better explaining the step in which this simulation occurs.
>
> > *There is a lack of comparison with other learned subsampling methods, notably adaptive in Van Gorp et al. (2021). While I appreciate that the policy is black-box, the end results seems to be a working subsampling mask for a given reconstruction model, and it would be very interesting to see how the derived masks work between different model archetypes.*
>
> We agree that it is interesting to compare the masks generated by a supervised approach to those generated by ADS, and have now added a detailed comparison with [3] to the Appendix. Our analysis compares masks generated to reconstruct MNIST, exploring the degree to which each method creates masks that are adapted to specific digits, along with reconstruction performance. In short, we found that the method in [3] produced less adaptive masks, possibly due to the fact that the sampler and reconstruction model are trained jointly.
>
> > *Figure 3 is quite hard to understand. Please add a textual description in the caption.*
>
> Figure 3 has now been updated to include specific steps labelled in the diagram and described in the caption. We hope that this enhances the clarity of the image and serves as a visual aid to the algorithm.
>
> > *I would be interested to see how much longer ADS takes compared to the baseline fixed subsampling due to the additional particles required.*
>
> We have now included in Appendix A.7 the execution time for ADS versus the fixed-mask approaches for the CelebA experiments, showing that ADS adds some increase in execution time which scales linearly with the number of measurements. We note also that while more compute is required for increasing numbers of particles, they can be computed in parallel, and should therefore be scalable given adequate compute.

---

> > ### Author Response · Authors · 2025-01-23
> > **Part 2**
> >
> > > *What is Np for the MRI experiment?*
> >
> > The choice for $N_p$ was 16 in the MRI experiment, as in the other experiments. We have now included this in the experiment description.
> >
> > > *Can the method be extended to finding common subsampling masks for multiple images? It would be interesting to see how the masks found by ADS differ and presumably degrade once a common mask for multiple images is deployed.*
> >
> > This is an interesting idea, and we believe that it can be implemented with a simple extension of ADS. In particular, we could track belief distributions for multiple images in parallel, and compute the average entropy over all distributions in order to identify a sampling region that would maximize information gain on average across all targets. As you point out, one might indeed expect the performance to degrade as the mask is averaged across more and more targets. This could be an interesting avenue for future work.
> >
> > > *There are some minor typos in the references, "{mri}", missing captialisation on names like "bayesian" etc.*
> >
> > Thank you for highlighting this, they have now been fixed.

---

> > > ### Comment · Reviewer_1Wyn · 2025-01-28
> > >
> > > I thank the authors for their extensive treatment of my comments and for the revisions to the manuscript. The additional experiments make the method more compelling.
> > >
> > > Minor comment: there are some broken reference links on pages 2 and 3.

---

> > > > ### Author Response · Authors · 2025-01-28
> > > >
> > > > We are pleased to hear that you have found the new experiments compelling and would like to thank you again for your excellent suggestions.
> > > >
> > > > Thank you also for flagging the broken reference links -- these have now been fixed in the latest revision.

---

### Review · Reviewer_LnKL · 2024-12-10

**Summary Of Contributions:**

### Summary of Contributions

This submission presents Active Diffusion Subsampling (ADS), a novel methodology for designing adaptive sampling strategies in inverse problems using diffusion models. The approach offers interpretability and task flexibility, surpassing fixed-mask baselines in performance while maintaining simplicity. The major contributions and new knowledge are as follows:

**1. Novel Algorithm for Active Subsampling**
- Introduces **Active Diffusion Subsampling (ADS)**, leveraging diffusion models to adaptively select subsampling masks.
- Employs **maximum entropy sampling** to actively minimize uncertainty in the reconstruction target \(x\) by choosing measurements \(y\) with the highest expected entropy.
- Operates using **pre-trained diffusion models**, removing the need for task-specific retraining.

**2. White-Box Policy Function**
- The sampling policy is derived from Bayesian Optimal Experimental Design principles:
  \[
  a^*_t = \arg \max_{a_t} \left[ H(y_t \mid A_t, y_{t-1}) \right],
  \]
  where \(H\) is entropy, ensuring maximal information gain.
- Provides transparency in decision-making, avoiding reliance on black-box methods common in prior approaches.

**3. Experimental Validation**
- **MNIST**: ADS achieves a significant reduction in Mean Absolute Error (MAE), outperforming fixed sampling baselines by factors exceeding \(2.5\times\) at low sampling rates.
- **CelebA**: Demonstrates improved reconstruction quality in Peak Signal-to-Noise Ratio (PSNR) and Structural Similarity Index (SSIM) compared to random and data variance strategies, particularly at medium sampling rates.
- **MRI Acceleration**: Competes with supervised sampling techniques, such as PG-MRI and SeqMRI, achieving a **91.26 SSIM** on the fastMRI dataset with only 25% \(k\)-space measurements.

**4. Computational Efficiency**
- Minimizes inference time by reusing quantities computed during the reverse diffusion process for mask selection.
- Adopts efficient particle-based modeling to represent posterior distributions, enabling computationally lightweight uncertainty quantification.

**5. Real-World Applicability**
- Validates performance in high-dimensional imaging tasks, with applications ranging from **image inpainting** to **MRI acceleration**, showcasing adaptability across domains.

**6. Broader Impact and Limitations**
- Reduces reliance on domain-specific expertise by automatically generating sampling strategies.
- Proposes directions for accelerating posterior sampling and extending ADS to batch sampling.

Experimental Findings
- ADS improves task-specific reconstruction performance without retraining, achieving **interpretable and adaptive sampling** policies.
- Sample-dependent mask designs result in significant improvements, especially in datasets like MNIST, where variance across samples is high.
- Offers insights into subsampling challenges at extremely low rates, proposing further investigations into diffusion model robustness in such settings.

ADS bridges theoretical contributions in Bayesian design with practical advances in adaptive subsampling, setting a new benchmark for efficient, interpretable sampling strategies in inverse problems.

**Audience:**

Yes

**Claims And Evidence:**

Yes

**Requested Changes:**

Recommendations for Improvement

1. **Address Sparse Data Challenges**:
   - Investigate improvements to posterior sampling under very low sampling rates, potentially incorporating prior knowledge or augmenting the reverse diffusion process.

2. **Optimize Inference Efficiency**:
   - To lower computational costs, explore accelerated inference strategies (quantization, distillation, or reduced diffusion steps).

3. **Expand Evaluation Scope**:
   - Validate ADS on non-image datasets or domains to demonstrate broader applicability.
   - Include tasks with diverse noise characteristics or measurement models to test robustness.

4. **Enhance Comparability**:
   - Benchmark against a wider range of supervised and unsupervised methods across different domains.

5. **Discuss Scalability**:
   - Address potential challenges in scaling ADS to high-dimensional data and outline strategies to mitigate these issues.

**Strengths And Weaknesses:**

Strengths

**1. Novelty**
- The introduction of **Active Diffusion Subsampling (ADS)** is a significant contribution, combining **diffusion models** with **Bayesian Optimal Experimental Design** to create adaptive subsampling strategies.
- The method addresses a critical gap by providing a **task-flexible and interpretable approach** to subsampling.

**2. Interpretability**
- Unlike black-box policy approaches, the **white-box entropy-based policy function** is grounded in Bayesian principles, enhancing transparency and interpretability of the sampling decisions.

**3. Flexibility**
- ADS is **model-agnostic** and does not require retraining for different tasks, making it broadly applicable across domains such as **image inpainting**, **MRI acceleration**, and more.

**4. Performance**
- The experimental results show that ADS significantly outperforms fixed-mask baselines and competes well with supervised sampling strategies in tasks like MRI acceleration.
- Strong results are reported for MNIST and CelebA datasets, where ADS achieves:
  - **Lower reconstruction error** (Mean Absolute Error, MAE).
  - **Higher quality metrics** (SSIM, PSNR) compared to baselines.

**5. Computational Efficiency**
- Efficient use of posterior samples and particle-based modeling minimizes the computational overhead during inference.
- Results demonstrate ADS can be computationally competitive even in resource-intensive applications like MRI acceleration.

**6. General Applicability**
- The submission highlights the potential of ADS for various domains by validating it on distinct datasets, suggesting strong generalization capabilities.

**7. Clear Presentation**
- The theoretical formulation and experimental results are well-documented, with detailed explanations of the methodology and intuitive visualizations to support claims (e.g., subsampling masks, reconstruction outputs).

Weaknesses

**1. Limited Improvement at Low Sampling Rates**
- ADS shows **diminished performance gains at extremely low subsampling rates** (e.g., < 5%), particularly in MRI acceleration.
  - Potential causes include limitations in posterior sampling accuracy with very few measurements.
  - Authors may need to explore techniques to enhance posterior estimation robustness under sparse data conditions.

**2. Inference Time**
- ADS relies on **diffusion posterior sampling**, which can be computationally expensive, especially with large reverse diffusion steps (e.g., 10k steps in fastMRI).
  - While competitive in execution time for MRI tasks, future work could explore **accelerated inference techniques**, such as:
    - Quantization.
    - Model distillation.
    - Batch sampling designs.

**3. Overlap with Fixed Strategies**
In cases like fastMRI, the designed masks show significant overlap with fixed-mask strategies, reducing ADS's relative performance advantage.
  - This may limit its novelty in scenarios where data distributions naturally favor fixed sampling patterns.

**4. Lack of Robustness Analysis Across Domains**
- The submission focuses heavily on image-based tasks. It would be beneficial to evaluate ADS on:
  - Non-image domains (e.g., time-series, tabular data).
  - Tasks with **different noise distributions** or measurement models.

**5. Dependency on Posterior Sampling**
- The accuracy of ADS hinges on the quality of posterior sampling achieved by the diffusion model. Limitations in posterior approximations (e.g., due to model capacity or noise) could undermine its performance.
  - Investigating **alternative posterior estimation techniques** may strengthen the method's robustness.

**6. Scalability to Higher Dimensions**
While results are promising for datasets like MNIST and CelebA, the scalability to higher-dimensional datasets (e.g., 4K medical images) remains unclear.
  - Discussing strategies for handling such data (e.g., leveraging parallelism or distributed computation) could enhance the method's appeal.

**7. Comparability with Other Methods**
- While ADS competes well with existing supervised methods, comparisons are limited to a few baselines. Incorporating additional benchmarks or broader state-of-the-art comparisons would strengthen the submission.

---

> ### Author Response · Authors · 2025-01-23
>
> Thank you for your constructive comments, highlighting opportunities for the paper to become more robust, and areas for future work. We address each comment in the following.
>
> > **1. Limited Improvement at Low Sampling Rates**
>
> This is an interesting phenomenon, and we agree that it is caused by a decreased accuracy in diffusion posterior sampling given very few measurements. We believe that while our work has highlighted this issue, and may be serve as a motivating example for the development of future posterior sampling techniques.
>
> > **2. Inference Time**
>
> We agree that employing further techniques to accelerate inference is an interesting direction for future work, and have discussed this in the *Limitations & Future Work* section.
>
> > **3. Overlap with Fixed Strategies**
>
> This is indeed an important point to consider when deciding whether it is beneficial to employ ADS in a given application. Through our experiments, we have shown that the benefits of active sampling are most significant when optimal masks vary widely across samples form the data distribution, and discussed this in our *Discussion* section.
>
> > **4. Lack of Robustness Analysis Across Domains**
>
> While most of our experiments are in the image domain, we would like to highlight that for our MRI experiment we trained a diffusion model on complex-valued data, and used a Fourier measurement model, which is distinct from the identity measurement model used for MNIST and CelebA. Additionally, we have now added a further experiment using ultrasound data, where our diffusion model is trained on beamformed ultrasound in the polar domain.
>
> > **5. Dependency on Posterior Sampling**
>
> While we do use a single posterior sampling algorithm (Diffusion Posterior Sampling) in this paper, we believe that the good reconstruction results across a variety of domains and data distributions provide evidence of the robustness of the method. We believe that exploring alternative posterior samplers is an interesting direction for future work, and have suggested this in our *Limitations and Future Work* section.
>
> > **6. Scalability to Higher Dimensions**
>
> In terms of dimensionality, ADS scales similarly to DPS, with the main difference being the additional entropy computation. The entropy computation scales primarily with the number of particles $N_p$ chosen to model the posterior distribution, and we therefore highlight this as a key scaling parameter for ADS. In light of this, we have included a new experiment in Appendix A.8 exploring how the reconstruction quality scales with $N_p$ for our CelebA experiments, and a new paragraph to the *Discussion* section discussing theoretical intuitions for choosing good values for $N_p$.
>
> > **7. Comparability with Other Methods**
>
> We have now included in Appendix A.4 a comparison to the supervised method *Active Deep Probabilistic Subsampling* (Van Gorp et al., 2021), including quantitative and qualitative analyses.

---

### Review · Reviewer_mJLx · 2025-01-14

**Summary Of Contributions:**

This paper proposes an interesting approach for optimizing the measurement mechanism for computational imaging systems using diffusion-based priors. The idea is that for a specific object $\mathbf{x}$, given a measurement operator and the corresponding measurements $\mathbf{y}$, if one is able to model the posterior $p(\mathbf{x} | \mathbf{y})$, then one can iteratively update A (by adding more sample locations) such that the mutual information between $\mathbf{x}$ and $\mathbf{y}$ is maximized.

**Audience:**

Yes

**Claims And Evidence:**

Yes

**Requested Changes:**

I would encourage the authors to better justify the practical relevance of this approach, ideally by adding discussion and/or numerical experiment for a case where such a scheme is known to be practicable via hardware design.

**Strengths And Weaknesses:**

**Strengths**
- The approach is novel, and the paper is well-written,
- The use of diffusion models in this context is interesting.

**Weaknesses**
I think the paper could do a better job of justifying why this scheme is practically relevant. For example, it is true that for MRI, the sampling masks are designed by domain experts keeping in mind the characteristics of typical objects as well as what is physically realizable in a scanner. However, I'm not aware of actual hardware where taking such adaptive samples is possible, for the experiments considered.

---

> ### Author Response · Authors · 2025-01-23
>
> Thank you for your review and suggestions, and for highlighting an important question about real-world applications of ADS.
>
> Firstly, we would like to mention that the primary motivation for choosing the fastMRI benchmark in this case was to facilitate a fair comparison with existing active sampling methods, since it is the most popular benchmark in the literature. Actively sampling MRI with current hardware is indeed limited by slew rates, which determine the length of time taken to move from one frequency in the k-space to another. Future work towards active sampling in MRI could look to extending ADS or other active sampling algorithms to include a further 'cost' associated with each measurement location determined by the length of time it would take to move to that frequency and make the acquisition. The model would then aim to get the best reconstruction possible in a given length of time, rather than with a given sampling budget.
>
> In order to provide a real-world example of ADS that is not hardware constrained, we have now included a new experiment demonstrating the use of ADS in ultrasound scan-line subsampling. In this application, the algorithm chooses which subset of focused beams to transmit in order to best reconstruct the fully-sampled beamformed image. Such a method might be employed to decrease the energy or bandwidth requirements for the probe, which is of interest for wireless probes, cloud-based ultrasound, and more. Flexible ultrasound transmit schemes are realizable on systems such as those developed by the company us4us [1]. We have also included a new paragraph in the introduction to provide further motivation by specifying some tasks in other fields where ADS may be useful, such as seismic and astronomical imaging.
>
> [1] https://us4us.eu/product/us4r/

---

### Author Response · Authors · 2025-01-23
**Summary of revisions**

We would like to thank all of the reviewers for their insightful comments and suggestions. We have now uploaded an updated version of the paper, with changes highlighted in orange. Our changes can be summarised as follows:
- Experimental results on the task of ultrasound scan-line subsampling in Section 5.4, providing a new real-world application of ADS.
- A comparison with the supervised method *Active Deep Probabilistic Subsampling*, including qualitative analyses of the masks generated, in Appendix A.4.
- Improved clarity in the exposition of the algorithm, through re-writing parts of Section 4.1, more comments in Algorithm 1, and improved labelling and captioning in Figure 3.
- New experiments measuring the runtime of ADS for different numbers of measurements in Appendix A.7, and the reconstruction quality gains from increasing the number of particles in Appendix A.8.

---

### Decision · Action_Editor_rvCk · 2025-02-20

**Recommendation:** Accept as is

**Comment:**

The authors have addressed all concerns raised by the reviewers. All reviewers recommend acceptance.

**Audience:**

Yes, for example, researchers in the diffusion community.

**Claims And Evidence:**

Evidence is provided for all claims.